# Curved neuromorphic image sensor array using a MoS₂-organic heterostructure inspired by the human visual recognition system

Changsoon Choi [1,2,7], Juyoung Leem [3,7], Minsung Kim [1,2,7], Amir Taqieddin [3], Chullhee Cho [3], Kyoung Won Cho[1,2], Gil Ju Lee [4], Hyojin Seung [1,2], Hyung Jong Bae [3], Young Min Song [4], Taeghwan Hyeon [1,2], Narayana R. Aluru [3], SungWoo Nam [3,5✉] & Dae-Hyeong Kim [1,2,6✉]

Conventional imaging and recognition systems require an extensive amount of data storage, pre-processing, and chip-to-chip communications as well as aberration-proof light focusing with multiple lenses for recognizing an object from massive optical inputs. This is because separate chips (*i.e.*, flat image sensor array, memory device, and CPU) in conjunction with complicated optics should capture, store, and process massive image information independently. In contrast, human vision employs a highly efficient imaging and recognition process. Here, inspired by the human visual recognition system, we present a novel imaging device for efficient image acquisition and data pre-processing by conferring the neuromorphic data processing function on a curved image sensor array. The curved neuromorphic image sensor array is based on a heterostructure of MoS₂ and poly(1,3,5-trimethyl-1,3,5-trivinyl cyclotrisiloxane). The curved neuromorphic image sensor array features photon-triggered synaptic plasticity owing to its quasi-linear time-dependent photocurrent generation and prolonged photocurrent decay, originated from charge trapping in the MoS₂-organic vertical stack. The curved neuromorphic image sensor array integrated with a plano-convex lens derives a pre-processed image from a set of noisy optical inputs without redundant data storage, processing, and communications as well as without complex optics. The proposed imaging device can substantially improve efficiency of the image acquisition and recognition process, a step forward to the next generation machine vision.

[1] Center for Nanoparticle Research, Institute for Basic Science (IBS), Seoul 08826, Republic of Korea. [2] School of Chemical and Biological Engineering, Institute of Chemical Processes, Seoul National University, Seoul 08826, Republic of Korea. [3] Department of Mechanical Science and Engineering, University of Illinois at Urbana-Champaign, Urbana, IL 61801, USA. [4] School of Electrical Engineering and Computer Science, Gwangju Institute of Science and Technology, Gwangju 61005, Republic of Korea. [5] Department of Materials Science and Engineering, University of Illinois at Urbana-Champaign, Urbana, IL 61801, USA. [6] Department of Materials Science and Engineering, Seoul National University, Seoul 08826, Republic of Korea. [7] These authors contributed equally: Changsoon Choi, Juyoung Leem, Minsung Kim. ✉email: swnam@illinois.edu; dkim98@snu.ac.kr

Advances in the imaging, data storage, and processing technology have enabled diverse image-data-based processing tasks[1]. The efficient acquisition and recognition of a target image is a key procedure in the machine vision applications[1,2], such as facial recognition[3] and object detection[2]. The image recognition based on conventional image sensors and data processing devices, however, is not ideal in terms of efficiency and power consumption[4,5], which are particularly important aspects for advanced mobile systems[6]. This is because conventional systems recognize objects through a series of iterative computing steps that require massive data storage, processing, and chip-to-chip communications[7,8] in conjunction with aberration-proof imaging with complicated optical components[9] (Supplementary Fig. 1a).

An example of the image recognition process based on the conventional system is shown in Supplementary Fig. 2. The raw image data over the entire time domain is obtained by a flat image sensor array with multi-lens optics[9,10] and stored in a memory device for frame-based image acquisition[1] (Supplementary Fig. 2a and b). The massive raw data is iteratively processed and stored by a central processing unit (CPU) and a memory device for event detection and data pre-processing[3] (Supplementary Fig. 2c and d). Then the pre-processed data is transferred to and processed by a graphics processing unit (GPU) based on a neural network algorithm[7] (e.g., vector-matrix multiplication[11]) for feature extraction and classification[12] (Supplementary Fig. 2e). Such iterative computing steps[1] as well as multi-lens optics[13] in the conventional imaging device increase system-level complexity.

In contrast, human vision outperforms conventional imaging and recognition systems particularly with regard to unstructured image classification and recognition[14]. In the human visual recognition system, visual scenes are focused by a single lens, detected by the hemispherical retina, transmitted to the brain through optic nerves, and recognized by a neural network in the visual cortex[1] (Supplementary Fig. 1b). The human-eye optics is much simpler than the multi-lens optics in conventional cameras[9,13]. A distinctive feature is extracted in the neural network from the visual information acquired by the human eye[15,16], which is used for image identification based on memories[17]. Therefore, the human visual recognition system can achieve higher efficiency than conventional image sensors and data processing devices[1].

Inspired by the neural network of the human brain, memristor crossbar arrays have been proposed for neuromorphic data processing[3,18], which can potentially replace the GPU[7,12]. Memristor arrays have demonstrated efficient vector-matrix multiplications by physically implementing the neural network in hardware[5,11]. However, these electrical neuromorphic devices cannot directly respond to optical stimuli, and thus still require separate image sensors, memory, and processors to capture, store, and pre-process the massive visual information, respectively[1] (Supplementary Fig. 1c). Meanwhile, inspired by the human eye, curved image sensor arrays have been proposed[19,20]. Although they could simplify the structure of the imaging module[10], they do not have data processing capabilities and thus additional processing units and memory modules are still needed.

Ideally, a novel imaging device inspired by the human visual recognition system, which enables neuromorphic image data pre-processing as well as simplified aberration-free image acquisition, is necessary to dramatically improve the efficiency of the imaging and recognition process for machine vision applications[1,21]. We herein present a curved neuromorphic image sensor array (cNISA) using a heterostructure of $MoS_2$ and poly(1,3,5-trimethyl-1,3,5-trivinyl cyclotrisiloxane) (pV3D3), aiming at aberration-free image acquisition and efficient data

pre-processing with a single integrated neuromorphic imaging device (Supplementary Fig. 1d). The cNISA integrated with a single plano-convex lens realizes unique features of the human visual recognition system, such as imaging with simple optics and data processing with photon-triggered synaptic plasticity. The cNISA derives a pre-processed image from a set of noisy optical inputs without repetitive data storage, processing, and communications as well as complicated optics, required in conventional imaging and recognition systems.

## Results

**Curved neuromorphic imaging device inspired by human vision.** Figure 1a, b shows schematic illustrations of the human visual recognition system and the curved neuromorphic imaging device, respectively. The human eye, despite its simple optics, enables the high-quality imaging without optical aberrations[22] because its hemispherical retina matches with the hemispherical focal plane of the single human-eye lens[9] (Fig. 1a). In addition, the neural network exhibits high efficiency for classification of unstructured data by deriving distinctive features of the input data based on the synaptic plasticity[14] (i.e., short-term plasticity (STP) and long-term potentiation (LTP); Fig. 1a inset); the intensity of the post-synaptic output signal is weighted by the frequency of pre-synaptic inputs[15].

All such efficient features of the human visual recognition system are incorporated into the curved neuromorphic imaging device. A single plano-convex lens focuses the incident light (e.g., massive noisy images) on cNISA that detects massive optical inputs and derives a pre-processed image through neuromorphic data processing (Fig. 1b). The concave curvature of cNISA matches with the Petzval surface of the lens, minimizing optical aberrations without the need of complicated multi-lens optics[13] (Supplementary Fig. 3). The photon-triggered electrical responses, which are similar to synaptic signals in the neural network, are enabled by the $MoS_2$-pV3D3 heterostructure and result in a weighted electrical output from optical inputs (Fig. 1b inset). Such an integrated imaging device enables the image acquisition and data pre-processing through a single readout operation.

The high efficiency of cNISA in comparison with conventional systems is explained in Fig. 1c, d. In case of a conventional imaging system with a conventional processor (i.e., von-Neumann architecture; Fig. 1c top), a flat image sensor array responds to incoming light (i.e., optical inputs) focused by multi-lens optics[10] and generates a photocurrent proportional to the intensity of applied optical inputs[23,24]. All measurements by the image sensor should be converted into digital signals and stored in a memory device for frame-based image acquisition[25]. Then massive electrical outputs (i.e., raw image data) are sent to a pre-processing module and processed iteratively[3]. The pre-processed data is sent to post-processing units (e.g., GPU) for additional processing and image recognition[12]. Meanwhile, neuromorphic chips have been proposed to overcome computational inefficiency of the conventional von-Neumann architecture[8] (Fig. 1c bottom). However, there are still inefficient aspects in terms of storage, transfer, and pre-processing of massive electrical outputs (i.e., raw image data) due to isolated construction of the image sensor array from the pre-processing units[1].

On the contrary, cNISA can simplify imaging and data pre-processing steps, and thereby maximize efficiency (Fig. 1d). The detailed description on the overall architecture for image acquisition, data pre-processing, and data post-processing is presented in Supplementary Fig. 4. The cNISA receives optical inputs through a single lens, which can simplify the optical system construction (Supplementary Fig. 3b). The detailed optical

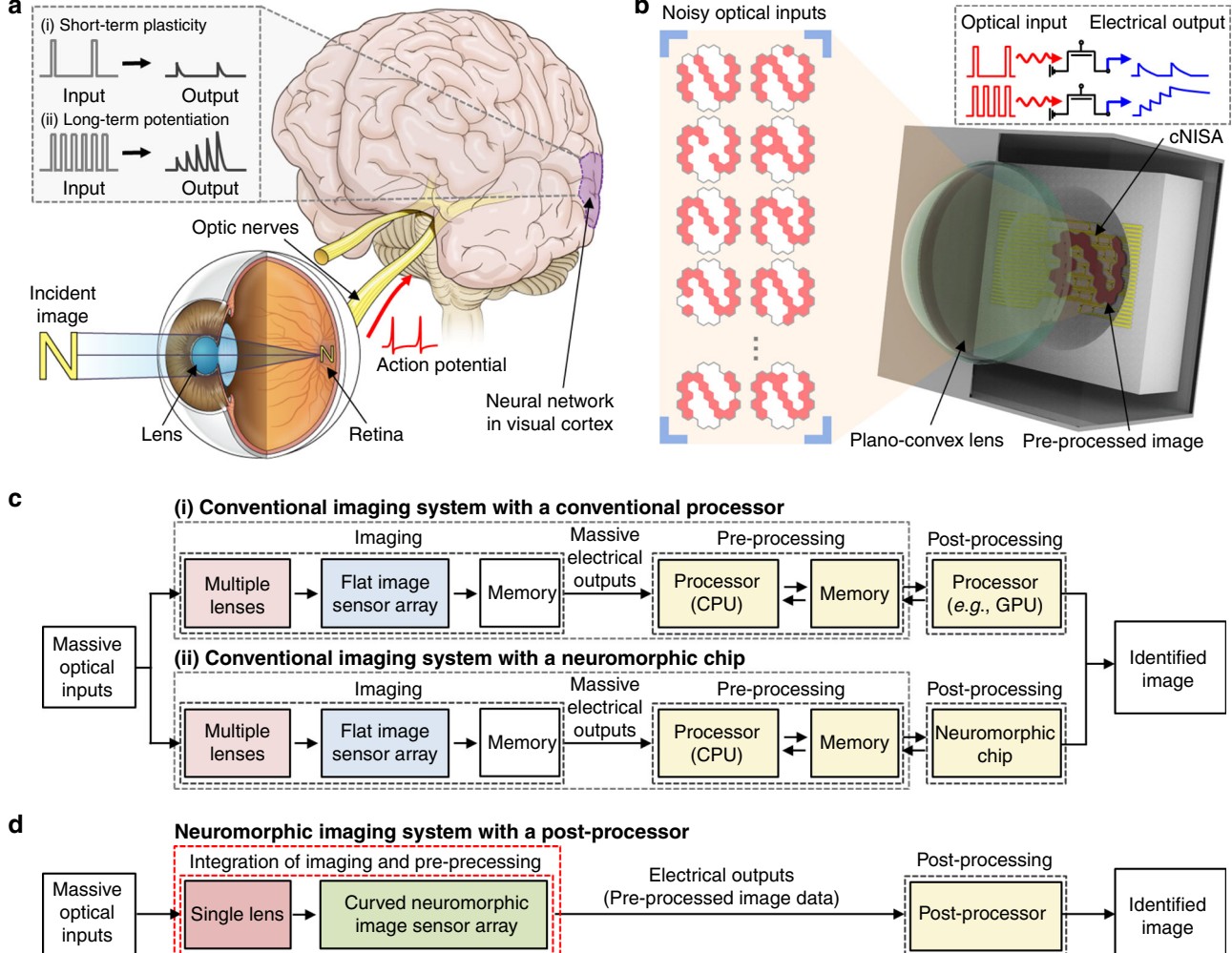

**Fig. 1 Curved neuromorphic imaging device inspired by the human visual recognition system. a** Schematic illustration of the human visual recognition system comprised of a single human-eye lens, a hemispherical retina, optic nerves, and a neural network in visual cortex. The inset schematic shows the synaptic plasticity (i.e., STP and LTP) of the neural network. **b** Schematic illustration of the curved neuromorphic imaging device. The inset in the dashed box shows the concept of photon-triggered synaptic plasticity that derives a weighted electrical output from massive optical inputs. **c** Block diagram showing the sequence of the image recognition using the conventional imaging and data processing system (e.g., conventional imaging system with a conventional processor (top) or with a neuromorphic chip (bottom)). **d** Block diagram showing the sequence of the image recognition using cNISA and a post-processor (e.g., GPU or neuromorphic chip).

analyses for the plano-convex lens and cNISA in comparison with the conventional imaging system are described in Supplementary Note 1 and Supplementary Tables 1 and 2. The output photocurrent gradually increases in a pixel with frequently repeated optical inputs (LTP; the bottom of Fig. 1b inset), while it decays in a pixel with infrequent optical inputs (STP; top of Fig. 1b inset). The electrical output at each pixel presents a weighted value proportional to the frequency of optical inputs, which includes the history of entire optical inputs. By a single readout of the electrical output, cNISA can derive a pre-processed image from a set of noisy optical inputs. Therefore, massive data storage, numerous data communications, and iterative data processing that have been required to obtain the pre-processed image data in conventional systems are not necessary[7,12].

**Photon-triggered synaptic plasticity**. The key principle of the neuromorphic imaging in cNISA is photon-triggered synaptic plasticity of the $MoS_2$-pV3D3 phototransistor (pV3D3-PTr). The pV3D3-PTr consists of a $Si_3N_4$ substrate, graphene source/drain electrodes, a $MoS_2$ light-sensitive channel, a pV3D3 dielectric layer,

and a Ti/Au gate electrode (Fig. 2a). An optical microscope image (top view) and cross-sectional transmission electron microscope (TEM) images are presented in Fig. 2b, c. The detailed synthesis, fabrication, and characterization procedures are described in Methods. The pV3D3-PTr exhibits the light-sensitive field-effect transistor behavior[26] and maintains its performance over three months (Supplementary Fig. 5).

The photo-response of pV3D3-PTr exhibits key characteristics of the synaptic plasticity in the human neural network. The photocurrent decays rapidly under infrequent optical inputs (e.g., two optical pulses with 10 s intervals; Fig. 2d), which corresponds to STP. However, the photocurrent is accumulated under frequent optical inputs (e.g., 20 optical pulses with 0.5 s intervals; Fig. 2e), which corresponds to LTP. In addition, the accumulated photocurrent becomes larger, as more frequent optical inputs are applied (Supplementary Fig. 6). Such a photon-triggered electrical response similar to the synaptic plasticity in the neural network is attributed to two characteristics of pV3D3-PTr, quasi-linear time-dependent photocurrent generation and prolonged photocurrent decay.

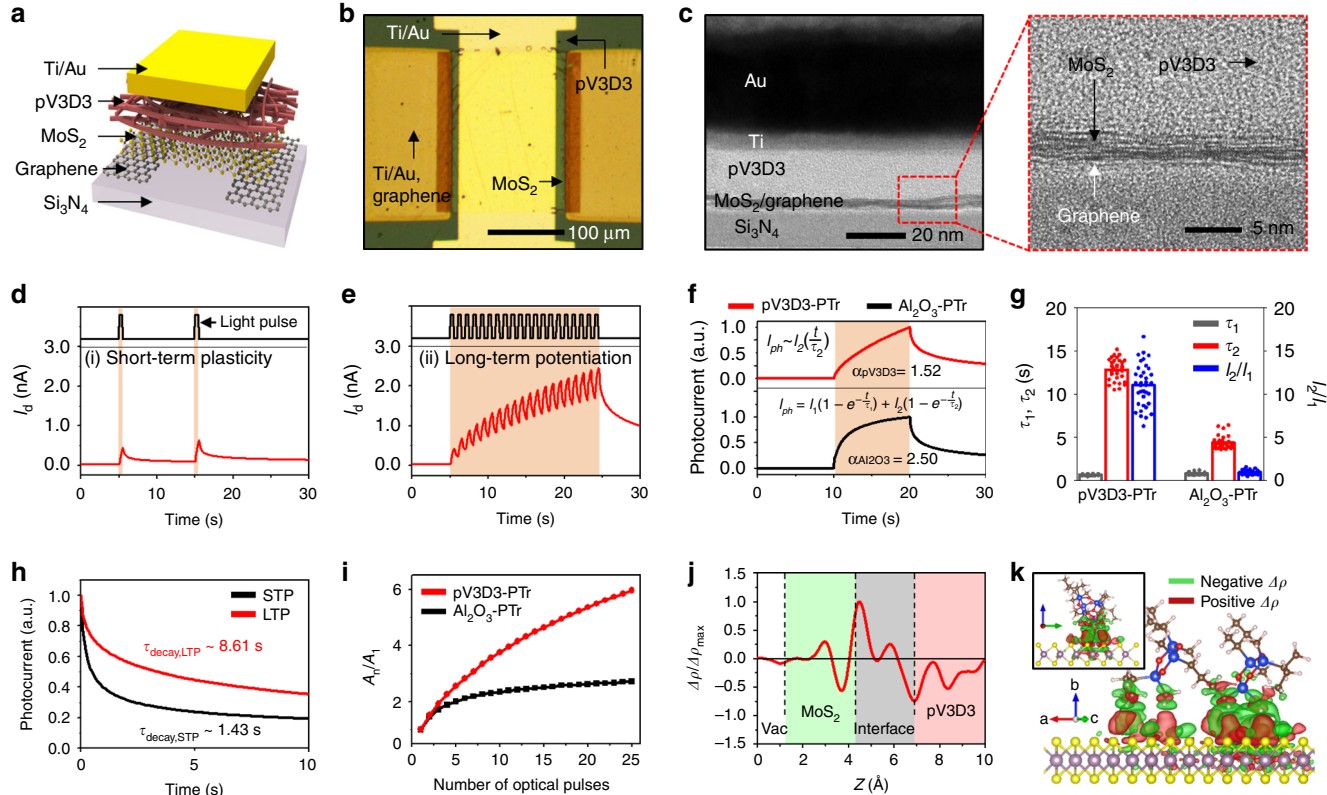

**Fig. 2 Photon-triggered neuromorphic behavior of the MoS$_2$-pV3D3 phototransistor. a** Schematic illustration of the device structure of pV3D3-PTr. **b** Optical microscope image of pV3D3-PTr. **c** Cross-sectional TEM images of pV3D3-PTr (left) and its magnified view (right). **d**, **e** Photon-triggered STP (**d**) and LTP (**e**) of pV3D3-PTr. **f** Photocurrent generation and decay characteristics of pV3D3-PTr and Al$_2$O$_3$-PTr. **g** Statistical analyses ($N = 36$) of time constants ($\tau_1$, $\tau_2$) and ratio of the photocurrent coefficient ($I_2/I_1$) for pV3D3-PTr and Al$_2$O$_3$-PTr. **h**, Photocurrent decay characteristics of pV3D3-PTr. For STP, a single optical pulse with 0.5 s duration was applied. For LTP, 20 optical pulses for 0.5 s duration each with 0.5 s intervals were applied. **i** $A_n/A_1$ of pV3D3-PTr and Al$_2$O$_3$-PTr as a function of the number of applied optical pulses. **j** Computationally obtained plane-averaged interfacial charge density difference in the MoS$_2$-pV3D3 heterostructure (i.e., $\Delta\rho = \rho_{MoS2,B} - \rho_{MoS2} - \rho_B$ where the subscript B indicates the dielectric) versus the distance in the aperiodic lattice direction. **k** Contour plots of the charge density difference in planes normal to the interface in the MoS$_2$-pV3D3 heterostructure. The green and red contours imply potential hole trapping and electron trapping sites, respectively. The inset shows a side view of Fig. 2k.

First, pV3D3-PTr exhibits the quasi-linear time-dependent photocurrent generation (Fig. 2f top). As a comparison, we prepared a control device, a MoS$_2$-Al$_2$O$_3$ phototransistor (Al$_2$O$_3$-PTr), which is a representative example of MoS$_2$-based phototransistors[9,27]. The Al$_2$O$_3$-PTr shows a non-linear time-dependent photocurrent generation behavior (Fig. 2f bottom). For quantitative comparison, a linearity factor ($\alpha$), a degree of linearity of the photocurrent increase with respect to the illumination time ($I_{ph}^{\alpha} \sim t$), is analyzed. As $\alpha$ approaches 1, the photocurrent increase becomes linear. However, if $\alpha$ is much larger than 1, the photocurrent increases nonlinearly and becomes saturated shortly, which hinders efficient pre-processing of data[1]. The linearity factor of pV3D3-PTr ($\alpha_{pV3D3}$) and that of Al$_2$O$_3$-PTr ($\alpha_{Al2O3}$) are obtained by fitting $\log(I_{ph})$ with respect to $\log(t)$, where $\alpha_{pV3D3}$ (1.52) is closer to unity than $\alpha_{Al2O3}$ (2.50). Therefore, pV3D3-PTr is more ideal for neuromorphic image data pre-processing than the control device (i.e., Al$_2$O$_3$-PTr)[28].

The time-dependent photocurrent generation of pV3D3-PTr in comparison with Al$_2$O$_3$-PTr is analyzed further by using an analytical model[29] (Supplementary Fig. 7). The details of the analytical model are described in Supplementary Note 2. The analytical model, $I_{ph}(t) = I_1(1-\exp(-t/\tau_1)) + I_2(1-\exp(-t/\tau_2))$, consists of two exponential photocurrent generation terms with time constants ($\tau_1$ and $\tau_2$) and photocurrent coefficients ($I_1$ and $I_2$). These parameters are compared in Fig. 2g. The pV3D3-PTr

exhibits a large photocurrent coefficient ratio ($I_{2,pV3D3}/I_{1,pV3D3} = 11.03$) and large $\tau_2$ ($\tau_{2,pV3D3} = 12.85$ s), resulting in a quasi-linear photocurrent generation function after series expansion of the exponential function ($I_{ph}(t) \cong I_{2,pV3D3}(t/\tau_{2,pV3D3})$; Supplementary Fig. 7a). In contrast, the control device (i.e., Al$_2$O$_3$-PTr) exhibits a much smaller photocurrent coefficient ratio ($I_{2,Al2O3}/I_{1,Al2O3} = 0.95$) and smaller $\tau_2$ ($\tau_{2,Al2O3} = 4.39$ s), thus showing non-linear photocurrent generation (Supplementary Fig. 7b).

Second, pV3D3-PTr shows prolonged photocurrent decay (Fig. 2h). The total decay time becomes longer with more frequent optical inputs. The decay time constant ($\tau_{decay}$), the time required for photocurrent decay to 1/e of an initial value, of pV3D3-PTr is dependent on the number of applied optical pulses (Supplementary Fig. 8). The decay time constant for LTP and STP ($\tau_{decay,LTP}$ and $\tau_{decay,STP}$) are 8.61 and 1.43 s, respectively (red line and black line in Fig. 2h), and the retention time for LTP and STP are 3600 and 1200 s, respectively (Supplementary Fig. 9). Such a large difference of the decay time and the retention time between LTP and STP are important to enhance the contrast of the pre-processed image.

With these attributes, pV3D3-PTr exhibits a high contrast between LTP and STP. The synaptic weight ($A_n/A_1$), a ratio of the photocurrent generated by n optical pulses ($A_n$) to the photocurrent generated by a single optical pulse ($A_1$), is defined to analyze the contrast quantitatively. The synaptic weight of pV3D3-PTr continuously increases as the number of applied

optical pulses increases (red line in Fig. 2i), whereas the synaptic weight of the control device (i.e., $Al_2O_3$-PTr) becomes almost saturated after five optical pulses (black line in Fig. 2i). Therefore, pV3D3-PTr exhibits a larger synaptic weight ($A_{25}/A_1$) of 5.93 than $Al_2O_3$-PTr with $A_{25}/A_1$ of 2.89 upon the irradiation of 25 optical pulses (Supplementary Fig. 10), leading to a better contrast in the neuromorphic imaging and pre-processing.

Such time-dependent photo-responses of $MoS_2$-based photo-transistors are related to charge trapping at the interface between $MoS_2$ and dielectrics[30]. Therefore, we theoretically investigated the interfacial properties of the $MoS_2$-pV3D3 heterostructure in comparison with those of the $MoS_2$-$Al_2O_3$ heterostructure. The $MoS_2$-pV3D3 heterostructure exhibits a large exciton binding energy (0.43 eV) that promotes spatial separation of electron-hole pairs at the interface[31,32]. In addition, its type-II band alignment and large conduction band energy difference facilitate the spatial charge separation[33,34] (Supplementary Fig. 11a). As a result, electrons and holes of a high density are spatially-separated and localized near the respective interface in the $MoS_2$-pV3D3 heterostructure (Fig. 2j). Such electron-hole pairs can induce photocurrent by the photogating effect[35]. The localized charges in the $MoS_2$-pV3D3 heterostructure were experimentally verified by the Kelvin probe force microscopy (Supplementary Fig. 12). Conversely, the charge separation at the $MoS_2$-$Al_2O_3$ hetero-structure is less probable (Supplementary Fig. 13a), because its type-I band alignment hinders the hole transfer at the interface[33] (Supplementary Fig. 11b). More details about the theoretical analysis are described in Supplementary Note 3.

The spatial distribution of the charge density difference ($\Delta\rho$), in which negative $\Delta\rho$ indicates the existence of potential hole trapping sites, was computationally analyzed. The $MoS_2$-pV3D3 heterostructure exhibits a spatially inhomogeneous distribution of $\Delta\rho$ in both in-plane and out-of-plane direction (Fig. 2k and its inset), compared to the relatively homogeneous distribution in the $MoS_2$-$Al_2O_3$ heterostructure (Supplementary Fig. 13b and its inset), due to the irregular geometry of the polymeric pV3D3 structure. Such an inhomogeneous distribution of $\Delta\rho$ results in the complex spatial and energy distribution of the potential hole trapping sites which are required for the active interfacial charge transfer[36].

**Image acquisition and neuromorphic data pre-processing**. The cNISA is applied to the acquisition of a pre-processed image from massive noisy optical inputs. First, a simplified version of the array ($3 \times 3$ array) was used to explain its operating principle. A set of noisy optical inputs ($I_m$), successively incident to the array, induces a weighted photocurrent ($I_{ph,n}$) in each pixel ($P_n$) (Fig. 3a). For example, $I_{ph,n}$ changes gradually by the irradiation of nine optical inputs ($I_1$–$I_9$; Supplementary Fig. 14). Pixels of #1, #2, #3, #5, and #8 receive eight optical pulses from nine incident inputs and thus generate a large accumulated photocurrent, whereas pixels of #4, #6, #7, and #9 receive only one optical pulse and thereby generate a negligible photocurrent (Fig. 3b). Since the final photocurrent of each pixel represents a weighted value, a pre-processed image can be obtained by simply mapping the final photocurrent.

The mapped images at different time points are shown in Fig. 3c. The image contrast is enhanced as more optical inputs are applied. Eventually, a pre-processed image 'T' is obtained from noisy optical inputs (Fig. 3d). Meanwhile, the pre-processed image is memorized and slowly dissipated over a long period of time (~30 s; Fig. 3e). These image pre-processing and signal dissipation processes resemble synaptic behaviors in the neural network (i.e., memorizing and forgetting)[14]. The remaining image can be immediately erased, if needed, by applying a positive gate

bias (e.g., $V_g = 1$ V) (Fig. 3f). The positive gate bias facilitates de-trapping of holes in the $MoS_2$-pV3D3 heterostructure, which removes the photogating effect and returns pV3D3-PTrs to the initial state[30]. Therefore, the subsequent image acquisition and pre-processing can be proceeded without interference by the afterimage in the previous imaging and pre-processing step[37].

For imaging demonstrations of more complex patterns, the array size is expanded from 9 pixels to 31 pixels, and other components are assembled for the integrated imaging system (Fig. 4a). The integrated system consists of a plano-convex lens that focuses incident optical inputs, cNISA that derives a pre-processed image from noisy optical inputs, and a housing that supports the lens and array (inset of Fig. 4a, b). The pixels in cNISA are distributed as a nearly circular shape (Supplementary Fig. 15). By employing an ultrathin device structure[38,39] (~2 μm thickness including encapsulations) and using intrinsically flexible materials (i.e., graphene[40], $MoS_2$[41–43], and pV3D3[44]), we could fabricate a mechanically deformable array. We also adopted a strain-releasing mesh design[45,46], added patterns to fragile materials (i.e., $Si_3N_4$), and located the array near the neutral mechanical plane. Therefore, the strain induced on the deformed array was <0.053 % (Supplementary Fig. 16). As a result, the array can be integrated on a concavely curved surface without mechanical failures (Fig. 4c). Additional details for mechanical analyses are described in Supplementary Note 4 and Supplementary Fig. 16d. A customized data acquisition system including current amplifiers and an analog-to-digital converter (ADC) enables the photocurrent measurement from cNISA (Fig. 4d and Supplementary Figs. 17 and 18). Each pixel of cNISA is serially connected to the current amplifier via an anisotropic conductive film (ACF). The detailed experimental setup and imaging procedures are described in Methods.

The image acquisition and pre-processing demonstrations by using the integrated system are shown in Fig. 4e–h. First, noisy optical inputs of C-shaped images (e.g., 20 optical inputs with 0.5 s durations and 0.5 s intervals; Supplementary Fig. 19a) are irradiated (Fig. 4e (i), red colored region). Then, a large accumulated photocurrent is generated in pixels with frequent optical inputs (Fig. 4g (i), red colored region), while a negligible photocurrent is generated in pixels with infrequent optical inputs (Fig. 4h (i), red colored region). As a result, a pre-processed image 'C' is acquired (Fig. 4f (i), red colored region). The pre-processed image is maintained approximately for 30 s (Fig. 4f–h (ii), blue colored region). The remaining image can be erased by applying a positive gate bias (e.g., $V_g = 1$ V; at 60 s of Fig. 4e–h (iii), violet colored region). Then, each pixel returns to the initial state (i.e., zero photocurrent). After erasing the afterimage, another imaging can be followed. For example, a pre-processed image 'N' is derived (Fig. 4e–h (iv), green colored region) from a set of noisy optical inputs (e.g., 20 images with 0.5 s durations and 0.5 s intervals; Supplementary Fig. 19b). These successful demonstrations of neuromorphic imaging, i.e., acquisition of a pre-processed image from massive noisy optical inputs through a single readout operation, exhibit the exceptional features of cNISA.

**Discussion**

In this study, we developed a curved neuromorphic imaging device by using a $MoS_2$-organic heterostructure. Distinctive features of the human visual recognition system, such as the curved retina with simple optics and the efficient data processing in the neural network with synaptic plasticity, have inspired the development of a novel imaging device for efficient image acquisition and pre-processing with the simple system construc-tion. The distinctive features of pV3D3-PTr, i.e., quasi-linear

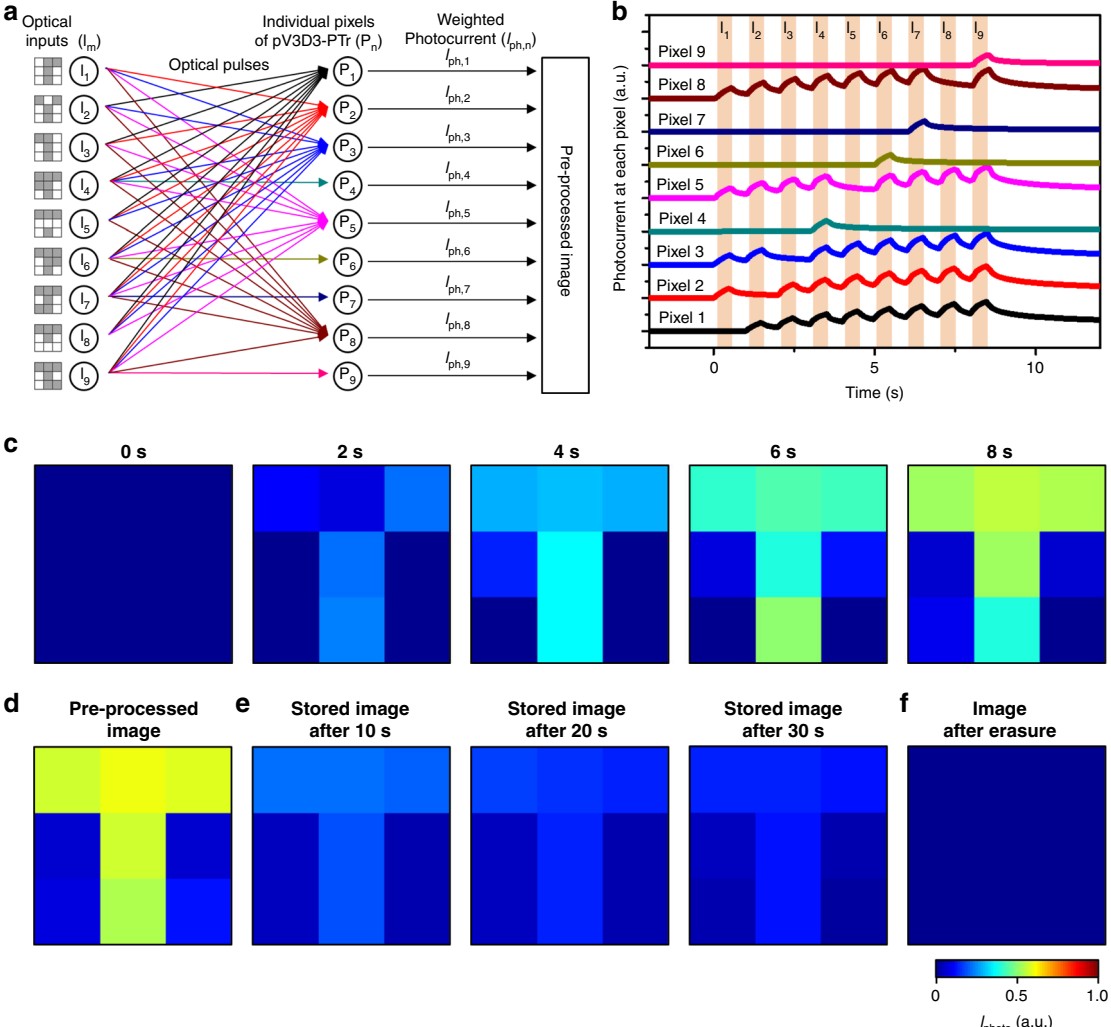

**Fig. 3 Operation principle of the neuromorphic imaging device. a** Schematic diagram showing the image acquisition and neuromorphic data pre-processing by using a 3 × 3 pV3D3-PTr array. **b** Normalized photocurrent measured at each pixel of the 3 × 3 pV3D3-PTr array. **c** Acquired image at each time point. **d** Pre-processed image obtained through image acquisition and neuromorphic data pre-processing. **e** Pre-processed image stored in the array while photocurrents of individual pixels slowly decay. **f** Erasure of the memorized image by applying a positive gate bias (i.e., $V_g = 1$ V).

time-dependent photocurrent generation and prolonged photocurrent decay characteristics, have enabled photon-triggered synaptic plasticity of cNISA. Such photon-triggered synaptic behaviors were analyzed by experimental studies and computational simulations. The imaging demonstrations using the integrated system proved that a pre-processed image can be successfully derived from a set of noisy optical inputs through a single readout operation.

Such a curved neuromorphic imaging device could enhance the efficiency for image acquisition and data pre-processing as well as simplify the optics to miniaturize the overall device size, and thus has a potential to be a key component for efficient machine vision applications. Note that detailed specifications of the curved neuromorphic imaging device are compared to those of the relevant neuromorphic image sensors in Supplementary Table 3. Nevertheless, additional processors for data post-processing, which extract features from the pre-processed image data and identify the target object, are still necessary for machine vision applications[47,48]. Therefore, further device units for efficient post-processing of the pre-processed image data should be still integrated[37], although the pre-processed image can be efficiently obtained by cNISA. Neuromorphic processors (e.g., memristor crossbar array) enable efficient post-processing of the

pre-processed image data in terms of the fast computation and the low power consumption[11]. The combination of cNISA with such neuromorphic processors would be helpful for demonstrating machine vision applications, although massive data storage and communications between them are still required due to their isolated architecture. In this regard, the development of a fully integrated system, which can perform the entire steps from image acquisition to data pre-/post-processing in a single device, can be an important goal in future research. The development of such technologies would make a step forward to the high-performance machine vision.

## Methods

**Synthesis of graphene and MoS₂**. Graphene was synthesized using chemical vapor deposition (CVD)[9]. After 30 min annealing of a copper foil (Alfa Aesar, USA) at 1000 °C under a constant flow of $H_2$ (8 sccm, 0.08 Torr), a flow of $CH_4$ (20 sccm, 1.6 Torr) was added for 20 min at 1000 °C. Then, the chamber was cooled down to room temperature under a flow of $H_2$ (8 sccm, 0.08 Torr).

$MoS_2$ was also synthesized using CVD[9]. Alumina crucibles with sulfur (0.1 g; Alfa Aesar, USA) and $MoO_3$ (0.3 g; Sigma Aldrich, USA) were placed at the upstream and center of the chamber, respectively. A $SiO_2$ wafer treated with the piranha solution and oxygen plasma was placed at the downstream of the $MoO_3$ crucible. After 30 min annealing at 150 °C under a constant flow of Ar (50 sccm, 10 Torr), the chamber was heated up to 650 °C for 20 min and maintained as 650 °C for 5 min under a constant flow of Ar (50 sccm, 10 Torr). The temperature

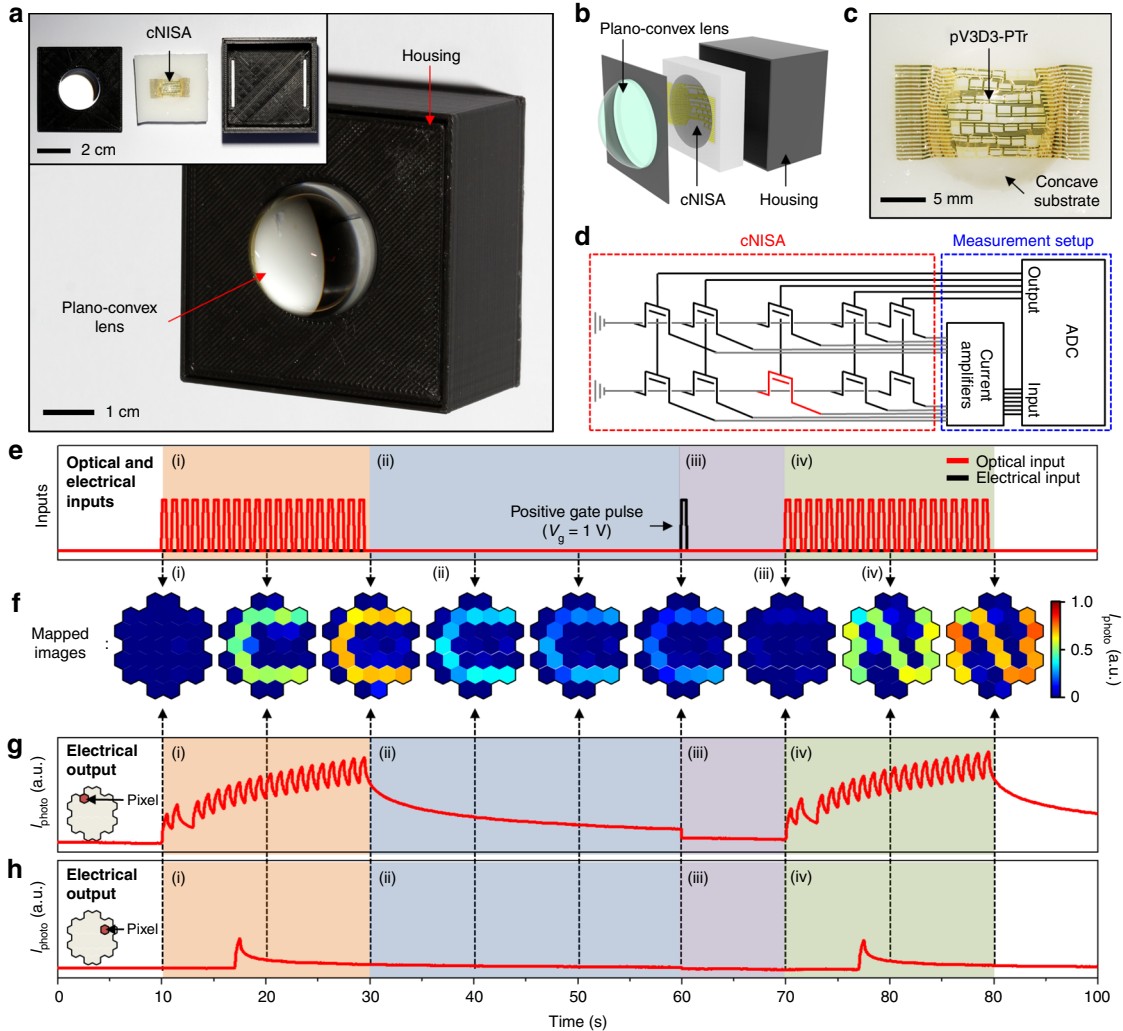

**Fig. 4 Fully integarted form of the curved neuromorphic imaging device. a** Photograph of an integrated imaging system that consists of a plano-convex lens, cNISA, and a housing. The inset shows the components before assembly. **b** Exploded illustration of the curved neuromorphic imaging device. **c** Photograph of cNISA on a concave substrate. **d** Schematic diagram of the customized data acquisition system for measuring the photocurrents of individual pixels in cNISA. **e–h** Demonstrations for deriving a pre-processed image from massive noisy optical inputs (e.g., acquisition of a pre-processed C-shape image (i), decay of the memorized C-shape image (ii), erasure of the afterimage (iii), and acquisition of a pre-processed N-shape image (iv)). Figure 4e shows applied optical inputs and an applied electrical input. Figure 4f shows obtained images at each time point. Figure 4g and h show the photocurrent obtained from the pointed pixels at each time point.

of sulfur was maintained as 160 °C. After the synthesis is finished, the chamber was naturally cooled down to room temperature under a constant flow of Ar (50 sccm, 10 Torr).

**Deposition of pV3D3**. pV3D3 was deposited using initiated CVD (iCVD)[44]. The sample was placed in the iCVD chamber, whose temperature was maintained as 50 °C. The temperature of the filament was maintained as 140 °C. During the synthesis, 1,3,5-trimethyl-1,3,5-trivinyl cyclotrisiloxane (V3D3, 95%; Gelest, USA) and di-tert-butyl peroxide (TBPO, 97%; Sigma Aldrich, USA) were vaporized and introduced to the chamber. The ratio of V3D3 and TBPO was controlled as 2:1, and the chamber pressure was maintained as 0.3 Torr.

**Fabrication of cNISA**. The fabrication of cNISA began with spin-coating of a polyimide (PI) film (~1 μm thickness, bottom encapsulation; Sigma Aldrich, USA) on a SiO$_2$ wafer. A thin layer of Si$_3$N$_4$ (~15 nm thickness, substrate) was deposited using plasma-enhanced CVD, and the Si$_3$N$_4$ film was etched into an island-shaped array using photolithography and dry etching. Graphene (~2 nm thickness, electrode) was transferred onto Si$_3$N$_4$. Thin Ti/Au layers (~5 nm/25 nm thickness) were deposited and used as an etch mask and probing pads. Graphene was patterned as interdigitated source/drain electrodes. An ultrathin MoS$_2$ layer (~4 nm thickness, light-sensitive channel) was transferred onto the graphene electrodes, and patterned by photolithography and dry etching. A pV3D3 layer (~25 nm thickness, dielectric) was deposited by iCVD, and was etched by photolithography

and dry etching as an island-shaped array. Then a lift-off process was used to pattern the Ti/Au layers (~5 nm/25 nm thickness, gate electrode) deposited by thermal evaporation. Ti/Au layers (~5 nm/25 nm thickness, ACF pad) were additionally deposited. Additional deposition of a parylene film (~1 μm thickness, top encapsulation) and dry etching completed fabrication of the neuromorphic image sensor array. The fabricated image sensor array was detached from the SiO$_2$ wafer using a water-soluble tape (3 M Corp., USA), and then transfer-printed onto a concave hemispherical substrate made of polydimethylsiloxane (PDMS; Dow Corning, USA). The radius of curvature and the subtended angle of the hemispherical substrate are 11.3 mm and 123.8°, respectively (Supplementary Fig. 16c). A housing for mechanically supporting the plano-convex lens and cNISA was fabricated using a 3D printer (DP200, prototech Inc., Republic of Korea).

**Electrical characterization of pV3D3-PTr**. The electrical properties of pV3D3-PTr were characterized by using a parameter analyzer (B1500A, Agilent, USA). A control device (i.e., Al$_2$O$_3$-PTr) was prepared for comparison by using a similar fabrication process with pV3D3-PTr. The Al$_2$O$_3$ layer (~25 nm) deposited at 200 °C through thermal atomic layer deposition was used as a dielectric layer for fabricating Al$_2$O$_3$-PTr[9]. A white light-emitting diode whose intensity is 0.202 mW cm$^{-2}$ was used as a light source for the device characterization. The emission spectrum of the white light-emitting diode is shown in Supplementary Fig. 20. The pulsed optical inputs with 0.5 s durations were programmed and generated by using the Arduino UNO.

**Characterization of a MoS₂-pV3D3 heterostructure**. High-resolution TEM images were obtained by using the Cs corrected TEM (JEM-ARM200F, JEOL, Japan) to analyze the vertical configuration of pV3D3-PTr. The surface potential of a MoS₂-pV3D3 heterostructure was measured by using the Kelvin probe force microscope (Dimension Icon, Bruker, USA). A control sample of a MoS₂-$Al_2O_3$ heterostructure was also prepared and characterized for comparison. Raman and photoluminescence (PL) spectra of the as-grown MoS₂, the MoS₂-pV3D3 heterostructure, and the MoS₂-$Al_2O_3$ heterostructure were analyzed by using Raman/PL micro-spectroscopy (Renishaw, Japan) with 532 nm laser. PL spectra were fitted by Gaussian function. More details about PL and Raman analyses are described in Supplementary Note 5 with Supplementary Figs. 21 and 22.

**Imaging demonstration**. The programmed white optical pulses with durations of 0.5 s, intervals of 0.5 s, and intensities of 0.202 mW cm$^{-2}$ were irradiated to cNISA for illumination of a series of 20 noisy optical inputs (Supplementary Fig. 19). The front aperture blocked the stray light, and the plano-convex lens focused the incoming light onto cNISA. The surface profile of the plano-convex lens is measured by a large-area aspheric 3D profiler (UA3P, Panasonic) (Supplementary Fig. 23), and the detailed information about the optical system is described in Supplementary Table 1. The cNISA received incident optical inputs and generated a weighted photocurrent. The current amplifiers (e.g., transimpedance amplifiers and inverters) and power supplying chips were assembled on a printed circuit board (PCB; Supplementary Fig. 18 and Supplementary Table 4). Each pixel in cNISA is individually connected to the current amplifier via ACF. The photocurrents of individual pixels were amplified by the current amplifiers and independently processed by ADC (USB-6289, National Instruments, USA).

## Data availability
The data files that support the findings of this study are available from the corresponding authors upon reasonable request.

## Code availability
We used Arduino 1.8.3 and LabVIEW 2017 to operate the custom-made circuits. The source codes for Arduino and LabVIEW are available from the corresponding authors upon reasonable request.

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

## Acknowledgements

This research was supported by IBS-R006-A1. S.N. acknowledges support from NSF (MRSEC DMR-1720633, ECCS-1935775, CMMI-1904216, and DMR-1708852), AFOSR (FA2386-17-1-4071), NASA ECF (NNX16AR56G), and ONR YIP (N00014-17-1-2830). N.R.A. acknowledges support from NSF (OISE-1545907, DMR-1708852, MRSEC DMR-1720633, and CMMI-1921578). A.T. and N.R.A. acknowledge the use of the parallel computing resources: (1) Blue Waters (supported by NSF awards OCI-0725070, ACI-1238993 and the state of Illinois, and as of December 2019, supported by the National Geospatial-Intelligence Agency), and (2) Comet at San Diego Supercomputer Center which is provided by the Extreme Science and Engineering Discovery Environment (XSEDE) (supported by National Science Foundation (NSF) Grant No. OCI1053575) under TG-CDA100010 allocation. C.C. acknowledges support from NASA Space Technology Research Fellow Grant No. 80NSSC17K0149.

## Author contributions

C.C., J.L., M.K., A.T., K.W.C., T.H., N.R.A., S.N., and D.-H.K. designed the experiments, analyzed the data and wrote the manuscript. C.C., M.K., and H.S. fabricated cNISA and performed the characterization of individual devices. C.C., J.L., A.T., C.C., H.J.B., and N.R.A. analyzed the $MoS_2$-pV3D3 heterostructure experimentally and theoretically. C.C. performed theoretical analysis on mechanics. G.J.L and Y.M.S. performed theoretical analysis on optics. All authors discussed the results and commented on the manuscript.

## Competing interests

The authors declare no competing interests.
