## [Peer Review File · Nature Communications]

REVIEWER COMMENTS

Reviewer #1 (Remarks to the Author):

The paper presents a Curved neuromorphic image sensor array taking the inspiration from the human visual recognition system. The strong points are:

They incorporate the biological vision system in the sensor, which is an essential topic in recent times.

1. The motivation for the work is well explained with a device-level fabrication.
2. The paper gives more emphasis on describing the sensor.
3. The PVT analysis is a strong point of the article.

The concerns about the paper are the following:

1. The authors mentioned in figure 1 that the design do pre-processing and post-processing from the raw data. However, they detail architectures for these processings are missing.
2. We can visualize that the overall processing is sequential, but our visual system performs the processing in parallel. How is the model brain-inspired?
3. The article is hard to comply with a complex organization. They present the figures after the reference and in a separate file also. This organization confuses readers. Besides, some pictures appear in the paper several times.
4. Their motivation is mainly adopting the visual recognition system in the sensor. It is desired to discuss the relevant works in literature and the performance comparison with them.
5. The authors miss referring to recent work such as "Bio-inspired smart vision sensor: toward a reconfigurable hardware modeling of the hierarchical processing in the brain. or "Event-Based Reconfigurable Hierarchical Processors for Smart Image Sensors." It would be interesting to observe the distinction among these works.

Reviewer #2 (Remarks to the Author):

Summary

The paper describes a concept and initial prototype for a combined visual imager and object recognition neural network based on the hardware implementation of a photon triggered MoS₂-pV3D3-PT_r artificial synapse. The paper is generally well written the figures clearly communicate the concepts and ideas. The paper merges material synthesis and characterization, electronic theory and modeling, and a prototype device. This work attempts to recreate certain characteristics of biological vision systems, including plasticity and potentiation using the optically stimulated field-effect transistor. Overall the paper is original work and demonstrates several potential advantages of thin-film electronics and 2D materials including the ability to be curved into 3D shapes to reduce optical complexity. The authors are commended in the extension of the excellent electronic materials development and modeling discoveries to a fully functional prototype imaging system. In the reported work, the sensitivity of the optical modulation is clearly visible and functionally, but the dynamics occur over the range of seconds, limiting practical applicability of this specific materials and phototransistor design. The paper presents an initial implementation of a physical neural network within this optical context; for comparison other groups focusing on crossbar networks for purely electrical signals (e.g. microphone outputs) have used hidden layers and training to perform very low power signal pattern recognition.

Questions for the authors:

1. More details should be provided on the optical experiments and experimental setup (Fig 4). For example, what is the lens type, focal length, radius of curvature, etc. Was any aperture used for the imaging testing?
2. Similarly, the curvature of the sensor is not reported, what is the radius of curvature and the subtended angle of curvature for the sensor layer?
3. Can the authors elaborate further on how the membrane sensor was formed without breakage was observed? Page 11 mentions a strain-releasing mesh design and refers to prior articles, but it is not clear

which components were patterned (i.e. MoS₂ , graphene, etc.) into a serpentine mesh, or if other features were included in the sensor device to accommodate the strains resulting from induced curvature.

4. The MoS₂ / pV3D3 clearly exhibits response to optical inputs ; however the presented experiments in Figure 2 do not provide complete details on the optical portion of the experiment . Can the authors provide the require optical illumination intensity (W/cm²) used in the experiments in Figs 3 and 4? What is the spectrum?

5. In figure 3, it is clear that there are no hidden layers that would store weighting functions to translate the built up potentiation to recognition of specific objects. It would be helpful if the authors could elaborate on the potential advantages or disadvantages to including additional processing layers in close proximity to the imaging surface.

Reviewer #1:

Summary Comments: The paper presents a Curved neuromorphic image sensor array taking the inspiration from the human visual recognition system. The strong points are:

They incorporate the biological vision system in the sensor, which is an essential topic in recent times.

1. The motivation for the work is well explained with a device-level fabrication.

2. The paper gives more emphasis on describing the sensor.

3. The PVT analysis is a strong point of the article.

The concerns about the paper are the following:

Our response to summary comments: We thank the reviewer for the detailed and insightful evaluation on our work. In the following, we responded to the reviewer's comments in a point-by-point manner.

Comment #1: The authors mentioned in figure 1 that the design does pre-processing and post-processing from the raw data. However, the detailed architectures for these processings are missing.

Our response to comment #1: We thank the reviewer for the comment. We added schematic illustrations for more detailed explanation on the architecture for image acquisition, data pre-processing, and data post-processing by using cNISA, a customized data acquisition system, and post-processors (*i.e.*, memory, CPU, and GPU) in the revised Supplementary Information.

Our modification to the manuscript:

(Line 13, page 6: in the revised main text)

“The detailed description on the overall architecture for image acquisition, data pre-processing, and data post-processing is presented in Supplementary Fig. 4.”

(Supplementary Figure 2: in the revised manuscript)

Supplementary Figure 2 | Image recognition process in conventional image sensors and data processing devices. Imaging and recognition steps, including image acquisition, event detection, data pre-processing, and data post-processing, in the conventional image sensors and data processing devices.

(Supplementary Figure 4: in the revised manuscript)

Supplementary Figure 4 | Overall architecture for imaging and recognition. Schematic diagram that describes the overall architecture for image acquisition, data pre-processing, and data post-processing by using cNISA, a customized data acquisition system, and post-processors.

Comment #2: We can visualize that the overall processing is sequential, but our visual system performs the processing in parallel. How is the model brain-inspired?

Our response to comment #2: We appreciate the reviewer’s comment. The human visual recognition system features efficient imaging and data processing, which are based on the synaptic plasticity of the neural network in the brain in addition to the single-lens-based imaging with the curved retina in the eye. Thus, we adopted these two features of the human visual recognition system into our imaging device to achieve the efficient image acquisition and data pre-processing in a single integrated device. Although overall processing in our system is still sequential if we consider the post-processing step, our device still has efficient aspects inspired from the human visual recognition system with regard to image acquisition and data pre-processing. We revised the manuscript to clarify these points.

Our modification to the manuscript:
(Line 1, page 4: in the revised main text)

“A distinctive feature is extracted in the neural network from the visual information acquired by the human eye^{15,16}, which is used for image identification based on memories¹⁷.”

(Line 22, page 4: in the revised main text)

“The cNISA integrated with a single plano-convex lens realizes unique features of the human visual recognition system, such as imaging with simple optics and data processing with photon-triggered synaptic plasticity.”

(Line 8, page 5: in the revised main text)

“In addition, the neural network exhibits high efficiency for classification of unstructured data by deriving distinctive features of the input data based on the synaptic plasticity¹⁴ (i.e., short-term plasticity (STP) and long-term potentiation (LTP); Fig. 1a inset); the intensity of the post-synaptic output signal is weighted by the frequency of pre-synaptic inputs¹⁵.”

(Line 18, page 5: in the revised main text)

“The photon-triggered electrical responses, which are similar to synaptic signals in the neural network, are enabled by the MoS₂-pV3D3 heterostructure and result in a weighted electrical output from optical inputs (Fig. 1b inset).”

(Line 2, page 13: in the revised main text)

“Distinctive features of the human visual recognition system, such as the curved retina with simple optics and the efficient data processing in the neural network with synaptic plasticity, have inspired the development of a novel imaging device for efficient image acquisition and pre-processing with the simple system construction.”

(Figure 1a: in the revised manuscript)

Figure 1a | Schematic illustration of the human visual recognition system comprised of a single human-eye lens, a hemispherical retina, optic nerves, and a neural network in visual cortex. The inset schematic shows the synaptic plasticity (*i.e.*, STP and LTP) of the neural network.

Comment #3: *The article is hard to comply with a complex organization. They present the figures after the reference and in a separate file also. This organization confuses readers. Besides, some pictures appear in the paper several times.*

Our response to comment #3: We thank the reviewer for the comment. In the original manuscript, we included figures with their corresponding captions once. But the same figures were automatically included at the end of manuscript again by the system of *Nature Communications*. This may have caused confusion to the reviewer, and we feel sorry for that. But this issue will be solved during the editing procedure. In addition, we revised the manuscript to avoid redundancy of some figure frames.

Our modification to the manuscript:
(Line 18, page 3: in the revised main text)

“Such iterative computing steps¹ as well as multi-lens optics¹³ (~~Supplementary Fig. 1a~~) in the conventional imaging device increase system-level complexity.”

(Line 24, page 5: in the revised main text)

“In case of a conventional imaging system with a conventional processor (*i.e.*, von-Neumann architecture; Fig. 1c top), a flat image sensor array responds to incoming light (*i.e.*, optical inputs) focused by multi-lens optics¹⁰ (~~Supplementary Fig. 3a~~) and generates a photocurrent proportional to the intensity of applied optical inputs²³.”

(Line 5, page 9: in the revised main text)

“The synaptic weight (A_n/A_1), a ratio of the photocurrent generated by n optical pulses (A_n) to the photocurrent generated by a single optical pulse (A_1), is defined to analyze the contrast quantitatively (~~Supplementary Fig. 9~~).”

(Line 2, page 12: in the revised main text)

“A customized data acquisition system including current amplifiers and an analog-to-digital converter (ADC) enables the photocurrent measurement from cNISA (Fig. 4d and ~~Supplementary Figs. 17 and 18~~).”

(~~Supplementary Figure 9: in the original manuscript; removed in the revised manuscript~~)

~~Supplementary Figure 9 | Definition of the synaptic weight. Photocurrent generation and decay characteristics of pV3D3 PTR upon the irradiation of 10 optical pulses. The synaptic weight is defined as A_n/A_1 .~~

(~~Supplementary Figure 17a: in the revised manuscript~~)

Supplementary Figure 17a | Exploded schematic illustration that shows the electrical connection between cNISA and the customized data acquisition system through the ACF.

Comment #4: Their motivation is mainly adopting the visual recognition system in the sensor. It is desired to discuss the relevant works in literature and the performance comparison with them.

Our response to comment #4: We appreciate the reviewer’s comment. As the reviewer pointed out, there have been innovative works for neuromorphic image sensors. We summarized the specifications of our curved neuromorphic imaging device and compared them to those of the relevant works.

Our modification to the manuscript:
(Line 12, page 13: in the revised main text)

“Such a curved neuromorphic imaging device could enhance the efficiency for image acquisition and data pre-processing as well as simplify the optics to miniaturize the overall device size, and thus has a potential to be a key component for efficient machine vision applications. Note that detailed specifications of the curved neuromorphic imaging device are compared to those of the relevant neuromorphic image sensors in Supplementary Table 3.”

(Line 2, page 21: in the revised main text)

“36. Jang, H. *et al.* An atomically thin optoelectronic machine vision processor. *Adv. Mater.* DOI://10.1002/adma.202002431 (2020).”

(Supplementary Table 3: in the revised manuscript)

Table 3 Specifications of neuromorphic image sensors					
Image sensor	Neuromorphic image sensor from Ref. 1	Neuromorphic image sensor from Ref. 14	Neuromorphic image sensor from Ref. 24	Neuromorphic image sensor from Ref. 36	Curved neuromorphic imaging device
Device type	Optoelectronic memory	Phototransistor	Photodiode	Phototransistor	Phototransistor
Materials	MoO _x	Amorphous IGZO	WSe ₂	MoS ₂	MoS ₂
Format	Array	Single device	Array	Array	Array
Optical inputs	UV	UV	Visible laser	Visible spectrum	Visible spectrum
Number of pixels	8×8	1	3×3	32×32	31
Optics	N.A.	N.A.	N.A.	N.A.	Single-lens optics (Plano-convex lens)
Applications	Image acquisition, pre-processing	N.A.	Pattern classification, autoencoding	Image acquisition, post-processing	Image acquisition, pre-processing
Reference	[1]	[14]	[24]	[36]	This work

Supplementary Table 3 | Specifications of neuromorphic image sensors. Comparison of the current curved neuromorphic imaging device with the state-of-the-art neuromorphic image sensors in terms of the device type, key materials, the device format, the type of optical inputs, the number of pixels, information for optics, and information for their applications.

Comment #5: *The authors miss referring to recent work such as "Bio-inspired smart vision sensor: toward a reconfigurable hardware modeling of the hierarchical processing in the brain. or "Event-Based Reconfigurable Hierarchical Processors for Smart Image Sensors." It would be interesting to observe the distinction among these works.*

Our response to comment #5: We thank the reviewer for the comment. The bio-inspired image sensors with reconfigurable hierarchical processors reported in the suggested references are highly promising options toward machine vision applications. We revised the manuscript to include these references and discuss the necessity of highly efficient post-processors for machine vision applications.

Our modification to the manuscript:
(Line 17, page 13: in the revised main text)

“Nevertheless, additional processors for data post-processing, which extract features from the pre-processed image data and identify the target object, are still necessary for machine vision applications^{46,47}. Therefore, further device units for efficient post-processing of the pre-processed image data should be still integrated³⁶, although the pre-processed image can be efficiently obtained by cNISA.”

(Line 23, page 21: in the revised main text)

“46. Bhowmik, P., Pantho, M. J. H. & Bobda, C. Bio-inspired smart vision sensor: toward a reconfigurable hardware modeling of the hierarchical processing in the brain. *J. Real-Time Image Proc.* DOI://10.1007/s11554-020-00960-5 (2020).

47. Bhowmik, P., Pantho, M. J. H. & Bobda, C. Event-based re-configurable hierarchical processors for smart image sensors. *2019 IEEE 30th International Conference on Application-specific Systems, Architectures and Processors* pp. 115-122. IEEE (2019).”

Thank you very much again for your insightful comments. We feel that these comments have helped to improve the quality of the manuscript significantly.

Reviewer #2:

Summary Comments: *The paper describes a concept and initial prototype for a combined visual imager and object recognition neural network based on the hardware implementation of a photon triggered MoS₂-pV3D3-PTTr artificial synapse. The paper is generally well written the figures clearly communicate the concepts and ideas. The paper merges material synthesis and characterization, electronic theory and modeling, and a prototype device. This work attempts to recreate certain characteristics of biological vision systems, including plasticity and potentiation using the optically stimulated field-effect transistor. Overall the paper is original work and demonstrates several potential advantages of thin-film electronics and 2D materials including the ability to be curved into 3D shapes to reduce optical complexity. The authors are commended in the extension of the excellent electronic materials development and modeling discoveries to a fully functional prototype imaging system., In the reported work, the sensitivity of the optical modulation is clearly visible and functionally, but the dynamics occur over the range of seconds, limiting practical applicability of this specific materials and phototransistor design. The paper presents an initial implementation of a physical neural network within this optical context; for comparison other groups focusing on crossbar networks for purely electrical signals (e.g. microphone outputs) have used hidden layers and training to perform very low power signal pattern recognition.*

Our response to summary comments: We sincerely appreciate the reviewer for precise comments. These comments were very helpful to improve the quality of our manuscript. The time-dependent photocurrent dynamics in our device confers the neuromorphic data processing function (*i.e.*, synaptic plasticity) on the image sensor, although there may be issues to be solved in the future. In this regard, we hope to emphasize that the main contribution of this work is in the development of a novel imaging device for efficient image acquisition and data pre-processing through a single readout operation in a simple miniaturized device. Previously, these processes (*i.e.*, image acquisition and data pre-processing) have required complicated optics as well as massive data storage, processing, and communications in multiple devices. Although future studies are needed for further integration and optimization for the post-processing part toward the practical applications, this device is expected to become a key component among various types of image sensors. We modified our manuscript to convey these points.

Our modification to the manuscript:

(Line 18, page 4: in the revised main text)

“We herein present a curved neuromorphic image sensor array (cNISA) using a heterostructure of MoS₂ and poly(1,3,5-trimethyl-1,3,5-trivinyl cyclotrisiloxane) (pV3D3), aiming at **aberration-free** image acquisition and **efficient** data pre-processing with a single integrated neuromorphic imaging device (Supplementary Fig. 1d).”

(Line 18, page 5: in the revised main text)

“The photon-triggered electrical responses, **which are similar to synaptic signals in the neural network, are enabled by the MoS₂-pV3D3 heterostructure and result** in a weighted electrical output from optical inputs (Fig. 1b inset).”

(Line 22, page 6: in the revised main text)

“By **a single readout of the** electrical output, cNISA can derive a pre-processed image from a set of noisy optical inputs. Therefore, massive data storage, numerous data communications, and iterative data

processing that have been required to obtain the pre-processed image data in conventional systems are not necessary^{7,12.}”

(Line 7, page 11: in the revised main text)

“The remaining image can be immediately erased, if needed, by applying a positive gate bias (*e.g.*, $V_g = 1$ V) (Fig. 3f). The positive gate bias facilitates de-trapping of holes in the MoS₂-pV3D3 heterostructure, which removes the photogating effect and returns pV3D3-PTrs to the initial state²⁹. Therefore, the subsequent image acquisition and pre-processing can be proceeded without interference by the afterimage in the previous imaging and pre-processing step³⁶.”

(Line 12, page 13: in the revised main text)

“Such a curved neuromorphic imaging device could enhance the efficiency for image acquisition and data pre-processing as well as simplify the optics to miniaturize the overall device size, and thus has a potential to be a key component for efficient machine vision applications.”

(Line 17, page 13: in the revised main text)

“Nevertheless, additional processors for data post-processing, which extract features from the pre-processed image data and identify the target object, are still necessary for machine vision applications^{46,47}. Therefore, further device units for efficient post-processing of the pre-processed image data should be still integrated³⁶, although the pre-processed image can be efficiently obtained by cNISA.”

(Line 2, page 21: in the revised main text)

“36. Jang, H. *et al.* An atomically thin optoelectronic machine vision processor. *Adv. Mater.* DOI://10.1002/adma.202002431 (2020).”

(Line 23, page 21: in the revised main text)

“46. Bhowmik, P., Pantho, M. J. H. & Bobda, C. Bio-inspired smart vision sensor: toward a reconfigurable hardware modeling of the hierarchical processing in the brain. *J. Real-Time Image Proc.* DOI://10.1007/s11554-020-00960-5 (2020).

47. Bhowmik, P., Pantho, M. J. H. & Bobda, C. Event-based re-configurable hierarchical processors for smart image sensors. *2019 IEEE 30th International Conference on Application-specific Systems, Architectures and Processors* pp. 115-122. IEEE (2019).”

Comment #1: More details should be provided on the optical experiments and experimental setup (Fig 4). For example, what is the lens type, focal length, radius of curvature, etc. Was any aperture used for the imaging testing?

Our response to comment #1: We thank the reviewer for the comment. We used a plano-convex lens whose radius of curvature is 13.127 mm. The focal length, indicating the distance between the lens and cNISA, was set to be 17.045 mm. In addition, a front aperture was used to block the stray light. We revised the original manuscript to provide details on the optical experiments and the experimental setup.

Our modification to the manuscript:

(Line 16, page 6: in the revised main text)

“The detailed optical analyses for the plano-convex lens and cNISA in comparison with the conventional imaging system are described in Supplementary Note 1 and Supplementary Tables 1 and 2.”

(Line 2, page 12: in the revised main text)

“A customized data acquisition system including current amplifiers and an analog-to-digital converter (ADC) enables the photocurrent measurement from cNISA (Fig. 4d and Supplementary Figs. 17 and 18).”

(Line 5, page 12: in the revised main text)

“The detailed experimental setup and imaging procedures are described in Methods.”

(Line 7, page 17: in the revised main text)

“The front aperture blocked the stray light, and the plano-convex lens focused the incoming light onto cNISA. The surface profile of the plano-convex lens is measured by a large-area aspheric 3D profiler (UA3P, Panasonic) (Supplementary Fig. 23), and the detailed information about the optical system is described in Supplementary Table 1.”

(Line 12, page 17: in the revised main text)

“The current amplifiers (e.g., transimpedance amplifiers and inverters) and power supplying chips were assembled on a printed circuit board (PCB; Supplementary Fig. 18 and Supplementary Table 4).”

(Supplementary Figure 17a: in the revised manuscript)

Supplementary Figure 17a | Exploded schematic illustration that shows the electrical connection between cNISA and the customized data acquisition system through ACF.

(Supplementary Figure 18: in the revised manuscript)

Supplementary Figure 18 | Layout of a customized data acquisition system. a,b, Layout of the front (a) and back (b) of the customized data acquisition system. U1-U31 indicate the current amplifiers (*i.e.*, transimpedance amplifiers and inverters), U32-U34 indicate the voltage regulators, and J1-J3 indicate the connections to external devices.

(Supplementary Figure 23: in the revised manuscript)

Supplementary Figure 23 | Surface profile of the plano-convex lens. a, Surface profile of the plano-convex lens. **b**, Surface roughness of the plano-convex lens. The best fit radius of curvature is 13.127178 mm, the RMS value of the roughness is 0.2798 μm , and the peak-to-peak variation of the roughness is 4.4593 μm .

(Supplementary Table 1: in the revised manuscript)

Table 1 Optics for curved neuromorphic imaging device				
Component	Radius (mm)	Thickness (mm)	Material	Semi-diameter (mm)
Object	Infinity	Infinity	-	Infinity
Aperture	Infinity	0.000	-	2.532
1	13.127	12.220	N-BK7	8.000
2	Infinity	17.045	-	3.653
Image sensor	-11.340	-	-	9.500

Supplementary Table 1 | Lens information of the curved neuromorphic imaging device. The radii, thicknesses, materials, and semi-diameters of the comprising lens component in the curved neuromorphic imaging device shown in Supplementary Fig. 3b.

(Supplementary Table 2: in the revised manuscript)

Table 2 Optics for conventional imaging device				
Component	Radius (mm)	Thickness (mm)	Material	Semi-diameter (mm)
Object	Infinity	Infinity	-	Infinity
1	34.333	4.435	N-BASF2	14.655
2	78.925	0.381	-	13.700
3	27.554	7.452	N-LAK8	12.424
4	592.999	2.032	SF2	9.879
5	16.807	7.036	-	7.669
Aperture	Infinity	9.276	-	4.465
7	-16.965	2.032	SF2	8.167
8	69.433	7.394	N-LAK33	10.825
9	-25.644	0.381	-	12.427
10	Infinity	5.989	N-LAK33	13.962
11	-58.641	0.381	-	14.888
12	79.263	3.348	N-LAK8	15.667
13	699.404	33.035	-	15.720
Image sensor	Infinity	-	-	30.309

Supplementary Table 2 | Lens information of the conventional imaging device. The radii, thicknesses, materials, and semi-diameters of the comprising lens components in the conventional imaging device shown in **Supplementary Fig. 3a**.

(Supplementary Table 4: in the revised manuscript)

Table 4 Components in customized data acquisition system			
Component	Type	Value	Model
U1-U31	Current amplifier (Transimpedance amplifier, Inverter)	N.A.	LMC662
U32	Voltage regulator	-2.0 V	LTC1550
U33	Voltage regulator	-5.0 V	LTC1983
U34	Voltage regulator	1.2 V	LTC3250
J1	Connection to ADC	N.A.	N.A.
J2, J3	ACF pad	N.A.	N.A.

Supplementary Table 4 | Components used in a customized data acquisition system. Detailed information of the components used in the customized data acquisition system shown in **Supplementary Fig. 18**.

Comment #2: Similarly, the curvature of the sensor is not reported, what is the radius of curvature and the subtended angle of curvature for the sensor layer?

Our response to comment #2: We thank the reviewer for the detailed comment. The radius of curvature and the subtended angle of cNISA are 11.3 mm and 123.8°, respectively. We revised the manuscript to include this information.

Our modification to the manuscript:
(Line 1, page 16: in the revised main text)

“The radius of curvature and the subtended angle of the hemispherical substrate are 11.3 mm and 123.8°, respectively (Supplementary Fig. 16c).”

(Supplementary Figure 16c: in the revised manuscript)

Supplementary Figure 16c | Schematic illustration of cNISA on the hemispherical substrate. Its radius of curvature is 11.3 mm, and its subtended angle is 123.8°.

Comment #3: Can the authors elaborate further on how the membrane sensor was formed without breakage was observed? Page 11 mentions a strain-releasing mesh design and refers to prior articles, but it is not clear which components were patterned (i.e. MoS₂, graphene, etc.) into a serpentine mesh, or if other features were included in the sensor device to accommodate the strains resulting from induced curvature.

Our response to comment #3: We thank the reviewer for the comment. We revised the manuscript to explain strategies for fabricating the concavely hemispherical image sensor array without mechanical fractures in more detail.

Our modification to the manuscript:
(Line 18, page 11: in the revised main text)

“By employing an ultrathin device structure^{37,38} (~2 μm thickness including encapsulations) and using intrinsically flexible materials (i.e., graphene³⁹, MoS₂⁴⁰⁻⁴², and pV3D3⁴³), we could fabricate a mechanically deformable array. We also adopted a strain-releasing mesh design^{44,45}, added patterns to fragile materials (i.e., Si₃N₄), and located the array near the neutral mechanical plane. Therefore, the strain induced on the deformed array was less than 0.053 % (Supplementary Fig. 16). As a result, the array can be integrated on a concavely curved surface without mechanical failures (Fig. 4c). Additional details for mechanical analyses are described in Supplementary Note 4 and Supplementary Fig. 16d.”

(Line 9, page 6: in the revised Supplementary Note 4)

“Supplementary Note 4. Mechanical analyses of the curved neuromorphic image sensor array
Finite element analysis for the strain distribution of cNISA on a hemispherical substrate was carried out by using COMSOL Multiphysics software (COMSOL Inc., USA). The mesh-patterned PI film whose thickness is 2 μm was deformed along the hemispherical substrate whose bending radius is 11.3 mm. It was assumed that the conformal contact was made. First principle mechanical strain with plasticity of PI was calculated by using an initial yield stress of 24.8 MPa, an isotropic tangent modulus of 1.39 GPa, and a Poisson’s ratio of 0.4.”

(Supplementary Figure 16a: in the revised manuscript)

Supplementary Figure 16a | Schematic illustration showing the design of the pV3D3-PT_r array.

(Supplementary Figure 16b: in the revised manuscript)

Supplementary Figure 16b | Schematic illustration showing an individual pixel of cNISA.

(Supplementary Figure 16d: in the revised manuscript)

Supplementary Figure 16d | First principal strain distribution calculation. The inset shows the first principal strain distribution in the individual pixel of cNISA.

Comment #4: The MoS₂/pV3D3 clearly exhibits response to optical inputs; however, the presented experiments in Figure 2 do not provide complete details on the optical portion of the experiment. Can the authors provide the require optical illumination intensity (W/cm²) used in the experiments in Figs 3 and 4? What is the spectrum?

Our response to comment #4: We appreciate the reviewer's comment. We added the normalized emission spectrum and the optical illumination intensity of the white light-emitting diode that we used in this work.

Our modification to the manuscript:
(Line 11 page 16: in the revised main text)

“A white light-emitting diode whose intensity is 0.202 mW cm^{-2} was used as a light source for the device characterization. The emission spectrum of the white light-emitting diode is shown in Supplementary Fig. 20.”

(Line 5, page 17: in the revised main text)

“The programmed white optical pulses with durations of 0.5 sec, intervals of 0.5 sec, and intensities of 0.202 mW cm^{-2} were irradiated to cNISA for illumination of a series of 20 noisy optical inputs (Supplementary Fig. 19).”

(Supplementary Figure 20: in the revised manuscript)

Supplementary Figure 20 | Characterization of a white light-emitting diode. Normalized emission spectrum of the white light-emitting diode which was used as a light source.

Comment #5: In figure 3, it is clear that there are no hidden layers that would store weighting functions to translate the built up potentiation to recognition of specific objects. It would be helpful if the authors could elaborate on the potential advantages or disadvantages to including additional processing layers in close proximity to the imaging surface.

Our response to comment #5: We thank the reviewer for the comment. As the reviewer pointed out, image recognition requires several steps including image acquisition, data pre-processing, and data post-processing. Although our curved neuromorphic imaging device enables efficient image acquisition and data pre-processing, additional processors for post-processing of the pre-processed data are still necessary for machine vision applications. We modified the manuscript to discuss these points including potential advantages and disadvantages.

Our modification to the manuscript:
(Line 21, page 13: in the revised main text)

“Neuromorphic processors (e.g., memristor crossbar array) enable efficient post-processing of the pre-processed image data in terms of the fast computation and the low power consumption¹¹. The combination of cNISA with such neuromorphic processors would be helpful for demonstrating machine vision

applications, although massive data storage and communications between them are still required due to their isolated architecture. In this regard, the development of a fully integrated system, which can perform the entire steps from image acquisition to data pre-/post-processing in a single device, can be an important goal in the future research. The development of such technologies would make a step forward to the high-performance machine vision.”

Thank you very much again for your insightful comments. We feel that these comments have helped to improve the quality of the manuscript significantly.

Other minor modifications:

#1 The order of authors was changed according to their contribution to the revision, one more affiliation of the corresponding author was added, and one more acknowledgement to the funding was added.

(Line 5, page 1: in the revised main text)

“Changsoon Choi^{1,2†}, Juyoung Leem^{3†}, Min Sung Kim^{1,2†}, Amir Taqieddin^{3,4}, **Chullhee Cho³**, Kyoung Won Cho^{1,2}, Gil Ju Lee⁵, Hyojin Seung^{1,2}, Hyung Jong Bae³, Young Min Song⁵, Taeghwan Hyeon^{1,2}, Narayana R. Aluru^{3,4}, SungWoo Nam^{3,6*}, and Dae-Hyeong Kim^{1,2,7*,**}”

(Line 21, page 1: in the revised main text)

“⁷Department of Materials Science and Engineering, Seoul National University, Seoul 08826, Republic of Korea.”

(Line 16, page 22: in the revised main text)

“C.C. acknowledges support from NASA Space Technology Research Fellow Grant No. 80NSSC17K0149.”

(Line 24, page 22: in the revised main text)

“C.C. performed theoretical analysis on mechanics.”

#2 Other minor modifications were made for further clarification.

(Line 16, page 2: in the revised main text)

“The cNISA integrated with a plano-convex lens derives a pre-processed image from a set of noisy optical inputs without redundant data storage, processing, and communications **as well as without complex optics**. The proposed imaging device can substantially improve efficiency of the image **acquisition and** recognition process, a step forward to the next generation machine vision.”

(Line 23, page 5: in the revised main text)

“The **high** efficiency of cNISA in comparison with conventional systems is explained in Figs. 1c and 1d.”

(Line 15, page 6: in the revised main text)

“The cNISA receives optical inputs through a single lens, which can simplify the **optical** system construction (Supplementary Fig. 3b).”

(Line 20, page 8: in the revised main text)

“The decay time constant (τ_{decay}), time required for photocurrent decay to 1/e of an initial value, of pV3D3-PTi is **dependent on** the number of applied optical pulses (Supplementary Fig. 8).”

(Line 9, page 10: in the revised main text)

“Such an inhomogeneous distribution of $\Delta\rho$ results in the complex spatial and energy distribution of the potential hole trapping sites which are required for the active interfacial charge transfer³⁵.”

(Figure 2g: in the revised manuscript)

Figure 2g | Statistical analyses ($N = 36$) of time constants (τ_1 , τ_2) and ratio of the photocurrent coefficient (I_2/I_1) for pV3D3-PTr and Al₂O₃-PTr.

REVIEWERS' COMMENTS

Reviewer #1 (Remarks to the Author):

My concerns have been acknowledged. Thank you for taking time to update the manuscript.

Reviewer #2 (Remarks to the Author):

Manuscript clarity and completeness have been significantly improved with the revision. The additional information and experimental details will help others in the field with understanding and benefiting from this work.

Reviewer #1:

Summary Comments: My concerns have been acknowledged. Thank you for taking time to update the manuscript.

Our response to summary comments: We sincerely appreciate the reviewer for evaluation on our revised work.

Thank you very much again for your insightful comments. We feel that these comments have helped to improve the quality of the manuscript significantly.

Reviewer #2:

Summary Comments: Manuscript clarity and completeness have been significantly improved with the revision. The additional information and experimental details will help others in the field with understanding and benefiting from this work.

Our response to summary comments: We thank the reviewer for positive evaluation on our revised work.

Thank you very much again for your insightful comments. We feel that these comments have helped to improve the quality of the manuscript significantly.

Modifications upon the editor’s request:

#1 Abstract was revised.

(Line 9, page 2: in the revised main text)

“Here, inspired by the human visual recognition system, we present a novel imaging device for efficient image acquisition and data pre-processing by conferring the neuromorphic data processing function on a curved image sensor array. The curved neuromorphic image sensor array (eNISA) is based on a heterostructure of MoS₂ and poly(1,3,5-trimethyl-1,3,5-trivinyl cyclotrisiloxane) (pV3D3). The curved neuromorphic image sensor array features photon-triggered synaptic plasticity owing to its quasi-linear time-dependent photocurrent generation and prolonged photocurrent decay, originated from charge trapping in the MoS₂-organic vertical stack. The curved neuromorphic image sensor array integrated with a plano-convex lens derives a pre-processed image from a set of noisy optical inputs without redundant data storage, processing, and communications as well as without complex optics.”

#2 The headings of each section and subsection were modified.

(Line 1, page 3: in the revised main text)

“**Introduction**”

(Line 5, page 4: in the revised main text)

“**Results**

Curved neuromorphic imaging device inspired by human vision”

(Line 5, page 7: in the revised main text)

“**Photon-triggered synaptic plasticity of the MoS₂-pV3D3 phototransistor”**

(Line 16, page 10: in the revised main text)

“**Image acquisition and neuromorphic data pre-processing”**

(Line 2, page 13: in the revised main text)

“**Discussion**”

#3 Mathematical terms were modified.

(Line 2, page 8: in the revised main text)

“For quantitative comparison, a linearity factor (α), a degree of linearity of the photocurrent increase with respect to the illumination time ($I_{ph}^{\alpha} \sim t$), is analyzed. As α approaches 1, the photocurrent increase becomes linear. However, if α is much larger than 1, the photocurrent increases nonlinearly and becomes saturated

shortly, which hinders efficient pre-processing of data¹. The linearity factor of pV3D3-PTr (α_{pV3D3}) and that of Al₂O₃-PTr (α_{Al2O3}) are obtained by fitting $\log(I_{ph})$ with respect to $\log(t)$, where α_{pV3D3} (1.52) is closer to unity than α_{Al2O3} (2.50).”

(Line 13, page 8: in the revised main text)

“The analytical model, $I_{ph}(t) = I_1(1-\exp(-t/\tau_1)) + I_2(1-\exp(-t/\tau_2))$, consists of two exponential photocurrent generation terms with time constants (τ_1 and τ_2) and photocurrent coefficients (I_1 and I_2).”

(Line 17, page 8: in the revised main text)

“The pV3D3-PTr exhibits a large photocurrent coefficient ratio ($I_{2,pV3D3}/I_{1,pV3D3} = 11.03$) and large τ_2 ($\tau_{2,pV3D3} = 12.85$ sec), resulting in a quasi-linear photocurrent generation function after series expansion of the exponential function ($I_{ph}(t) \cong I_{2,pV3D3}(t/\tau_{2,pV3D3})$; Supplementary Fig. 7a). In contrast, the control device (*i.e.*, Al₂O₃-PTr) exhibits a much smaller photocurrent coefficient ratio ($I_{2,Al2O3}/I_{1,Al2O3} = 0.95$) and smaller τ_2 ($\tau_{2,Al2O3} = 4.39$ sec), thus showing non-linear photocurrent generation (Supplementary Fig. 7b).”

(Line 22, page 8: in the revised main text)

“The total decay time becomes longer with more frequent optical inputs. The decay time constant (τ_{decay}), time required for photocurrent decay to 1/e of an initial value, of pV3D3-PTr is dependent on the number of applied optical pulses (Supplementary Fig. 8). The decay time constant for LTP and STP ($\tau_{decay,LTP}$ and $\tau_{decay,STP}$) are 8.61 sec and 1.43 sec, respectively (red line and black line in Fig. 2h), and the retention time for LTP and STP are 3,600 sec and 1,200 sec, respectively (Supplementary Fig. 9).”

(Line 7, page 9: in the revised main text)

“The synaptic weight (A_n/A_1), a ratio of the photocurrent generated by n optical pulses (A_n) to the photocurrent generated by a single optical pulse (A_1), is defined to analyze the contrast quantitatively.”

(Line 13, page 9: in the revised main text)

“Therefore, pV3D3-PTr exhibits a larger synaptic weight (A_{25}/A_1) of 5.93 than Al₂O₃-PTr with A_{25}/A_1 of 2.89 upon the irradiation of 25 optical pulses (Supplementary Fig. 10), leading to a better contrast in the neuromorphic imaging and pre-processing.”

(Line 7, page 10: in the revised main text)

“The spatial distribution of the charge density difference ($\Delta\rho$), in which negative $\Delta\rho$ indicates existence of potential hole trapping sites, was computationally analyzed. The MoS₂-pV3D3 heterostructure exhibits a spatially inhomogeneous distribution of $\Delta\rho$ in the both in-plane and out-of-plane direction (Fig. 2k and its inset), compared to the relatively homogeneous distribution in the MoS₂-Al₂O₃ heterostructure (Supplementary Fig. 13b and its inset), due to the irregular geometry of the polymeric pV3D3 structure. Such an inhomogeneous distribution of $\Delta\rho$ results in the complex spatial and energy distribution of the potential hole trapping sites which are required for the active interfacial charge transfer³⁵.”

(Line 19, page 10: in the revised main text)

“A set of noisy optical inputs (I_m), successively incident to the array, induces a weighted photocurrent ($I_{ph,n}$)

in each pixel (P_n) (Fig. 3a). For example, $I_{ph,n}$ changes gradually by the irradiation of nine optical inputs (I_1 - I_9 ; Supplementary Fig. 14).”

(Line 9, page 11: in the revised main text)

“The remaining image can be immediately erased, if needed, by applying a positive gate bias (e.g., $V_g = 1$ V) (Fig. 3f).”

(Line 16, page 12: in the revised main text)

“The remaining image can be erased by applying a positive gate bias (e.g., $V_g = 1$ V; at 60 sec of Figs. 4e-4h (iii), violet colored region).”

(Line 1, page 3: in the revised Supplementary Note 2)

“The analytical model is given as:

$$I_{ph}(t) = I_1(1-\exp(-t/\tau_1)) + I_2(1-\exp(-t/\tau_2))$$

where τ_1 , τ_2 are time constants, I_1 , I_2 are coefficients of the photocurrent, and t is the time for light irradiation².

Based on the photocurrent measurement data in Fig. 2f, the time constants (τ_1 , τ_2) and the ratio of photocurrent coefficients (I_2/I_1) of pV3D3-PTr and Al₂O₃-PTr were estimated. The estimated τ_1 , τ_2 , and I_2/I_1 of pV3D3-PTr are 0.68 sec, 12.85 sec, and 11.03, respectively, and those of Al₂O₃-PTr are 0.80 sec, 4.39 sec, and 0.95, respectively.

Using these estimated parameters, we can analyze the contribution from each photocurrent term in the model to the overall photocurrent, i.e., the relative contribution by $I_{ph,1}(t)$, i.e., $I_1(1-\exp(-t/\tau_1))$, and $I_{ph,2}(t)$, i.e., $I_2(1-\exp(-t/\tau_2))$, to $I_{ph}(t)$.”

(Line 15, page 3: in the revised Supplementary Note 2)

“ $I_{ph,1}(t)$ increases rapidly but becomes saturated shortly (blue dotted line in Supplementary Figs. 7a and 7b). In contrast, $I_{ph,2}(t)$ increases quasi-linearly without saturation (red dotted line in Supplementary Figs. 7a and 7b).”

(Line 18, page 3: in the revised Supplementary Note 2)

“For example, if the contribution of $I_{ph,2}(t)$ is dominant, the overall photocurrent increases quasi-linearly.”

(Line 21, page 3: in the revised Supplementary Note 2)

“In case of pV3D3-PTr, $I_{2,pV3D3}$ is an order of magnitude larger than $I_{1,pV3D3}$ ($I_{2,pV3D3}/I_{1,pV3D3} \sim 11.03$). In addition, $\tau_{2,pV3D3}$ is 18.90 times larger than $\tau_{1,pV3D3}$. As a result, $I_{ph,2}(t)$ is much larger than $I_{ph,1}(t)$, which leads to the quasi-linear increase of the overall photocurrent (Supplementary Fig. 7a). And, the overall photocurrent can be approximated as $I_{ph}(t) \cong I_{2,pV3D3}(t/\tau_{2,pV3D3})$ for small t (i.e., $t/\tau_{2,pV3D3} \ll 1$).”

(Line 3, page 4: in the revised Supplementary Note 2)

“First, $I_{2,Al2O3}$ and $I_{1,Al2O3}$ are comparable ($I_{2,Al2O3}/I_{1,Al2O3} \sim 0.95$). Furthermore, $\tau_{2,Al2O3}$ (4.39 sec) is smaller than $\tau_{2,pV3D3}$ (12.85 sec).”

(Line 22, page 5: in the revised Supplementary Note 3)

“The interfacial charge density, $\Delta\rho = \rho_{\text{MoS}_2, \text{B}} - \rho_{\text{MoS}_2} - \rho_{\text{B}}$ where the subscript B is either dielectric (e.g., pV3D3 or Al_2O_3), was also computed to investigate the charge density distribution and potential charge trapping sites at the interface of MoS_2 -pV3D3 heterostructure and MoS_2 - Al_2O_3 heterostructure.”

(Line 2, page 5: in the revised Supplementary Note 5)

“Although PL intensity of the MoS_2 -pV3D3 heterostructure is consistent with that of the as-grown MoS_2 , the intensity ratio between charged exciton and A exciton ($I_{\text{A}}/I_{\text{A}}$) increases by the pV3D3 deposition, which suggests small amount of electron doping effect by the deposited pV3D3 layer¹³.”

(Figure 2d-2k: in the revised manuscript)

Figure 2d-2k | **d,e**, Photon-triggered STP (**d**) and LTP (**e**) of pV3D3-PTr. **f**, Photocurrent generation and decay characteristics of pV3D3-PTr and Al_2O_3 -PTr. **g**, Statistical analyses ($N = 36$) of time constants (τ_1 , τ_2) and ratio of the photocurrent coefficient (I_2/I_1) for pV3D3-PTr and Al_2O_3 -PTr. **h**, Photocurrent decay characteristics of pV3D3-PTr. For STP, a single optical pulse with 0.5 sec duration was applied. For LTP, 20 optical pulses for 0.5 sec duration each with 0.5 sec intervals were applied. **i**, A_n/A_1 of pV3D3-PTr and Al_2O_3 -PTr as a function of the number of applied optical pulses. **j**, Computationally obtained plane-averaged interfacial charge density difference in the MoS_2 -pV3D3 heterostructure (i.e., $\Delta\rho = \rho_{\text{MoS}_2, \text{B}} - \rho_{\text{MoS}_2} - \rho_{\text{B}}$ where the subscript B indicates the dielectric) versus the distance in the aperiodic lattice direction. **k**, Contour plots of the charge density difference in planes normal to the interface in the MoS_2 -pV3D3 heterostructure. The green and red contours imply potential hole trapping and electron trapping sites, respectively. The inset shows a side view of Fig. 2k.

(Figure 3a: in the revised manuscript)

Figure 3a | Schematic diagram showing the image acquisition and neuromorphic data pre-processing by using a 3×3 pV3D3-PTr array.

(Figure 3f: in the revised manuscript)

Figure 3f | Erasure of the memorized image by applying a positive gate bias (*i.e.*, $V_g = 1$ V).

(Figure 4e-4h: in the revised manuscript)

Figure 4e-4h | e-h, Demonstrations for deriving a pre-processed image from massive noisy optical inputs (*e.g.*, acquisition of a pre-processed C-shape image (i), decay of the memorized C-shape image (ii), erasure of the afterimage (iii), and acquisition of a pre-processed N-shape image (iv)). Figure 4e shows applied optical inputs and an applied electrical input. Figure 4f shows obtained images at each time point. Figures 4g and 4h show the photocurrent obtained from the pointed pixels at each time point.

(Supplementary Figure 5: in the revised manuscript)

Supplementary Figure 5 | Time-dependent electrical characterization of pV3D3-PTr. **a**, Transfer curves of as-fabricated pV3D3-PTr under the light-off and light-on condition. **b**, Transfer curves of pV3D3-PTr stored in the ambient condition for three months. Its transfer characteristics are consistent with the original transfer curves.

(Supplementary Figure 6: in the revised manuscript)

Supplementary Figure 6 | Photocurrent generation and decaying characteristics of pV3D3-PTr for the optical inputs of various frequencies. Photocurrent generation and decaying characteristics of pV3D3-PTr in response to a series of optical inputs with various time intervals.

(Supplementary Figure 7: in the revised manuscript)

Supplementary Figure 7 | Modeling of the time-dependent photocurrent generation. **a,b**, Photocurrent generation characteristics of pV3D3-PTr (**a**) and Al₂O₃-PTr (**b**). The black line shows the overall photocurrent, and the blue and red dotted lines show the individual photocurrent terms in the analytical model. (i.e., $I_{ph}(t) = I_{ph,1}(t) + I_{ph,2}(t)$ in which $I_{ph,1}(t) = I_1(1 - \exp(-t/\tau_1))$ and $I_{ph,2}(t) = I_2(1 - \exp(-t/\tau_2))$).

(Supplementary Figure 8: in the revised manuscript)

Supplementary Figure 8 | Prolonged photocurrent decay with more optical inputs. The decay time constant (τ_{decay}) of pV3D3-PTr increases as the number of applied optical pulses increases.

(Supplementary Figure 9: in the revised manuscript)

Supplementary Figure 9 | Retention characteristics. **a**, Photocurrent decay characteristics of pV3D3-PTr in response to a single optical pulse with 0.5 sec duration. **b**, Photocurrent decay characteristic of pV3D3-PTr in response to 20 optical pulses with 0.5 sec durations and 0.5 sec intervals.

(Supplementary Figure 10: in the revised manuscript)

Supplementary Figure 10 | Synaptic weight of pV3D3-PTr and Al₂O₃-PTr under 25 optical pulses. **a,b**, Photocurrent generation and decaying characteristics of pV3D3-PTr (**a**) and Al₂O₃-PTr (**b**) upon the irradiation of 25 optical pulses. Synaptic weights can be calculated as A_{25}/A_1 .

(Supplementary Figure 12: in the revised manuscript)

Supplementary Figure 12 | Kelvin probe force microscopy measurement. Surface potential measurement of the MoS₂-pV3D3 heterostructure. The light condition was changed from off-condition (region 1) to on-condition (region 2) during the scanning (*i.e.*, light was turned on during the measurement). The black arrow at the bottom indicates the scanning direction.

(Supplementary Figure 13: in the revised manuscript)

Supplementary Figure 13 | Computed interfacial charge density of the MoS₂-Al₂O₃ heterostructure. **a**, Computed interfacial charge density in the MoS₂-Al₂O₃ heterostructure (*i.e.*, $\Delta\rho$) depending on the distance in the aperiodic lattice direction. The charge density is computed using the DFT-D3 method. **b**, Contour plots of the charge density difference in planes normal to the interface in the MoS₂-Al₂O₃ heterostructure. The green and red contours imply potential hole trapping and electron trapping sites, respectively. The inset shows a side view of Supplementary Fig. 13b.

(Supplementary Figure 17c: in the revised manuscript)

Supplementary Figure 17c | Schematic diagram of the current amplifier used in the customized data acquisition system for translating the photocurrent into amplified voltage signals.

(Supplementary Figure 21: in the revised manuscript)

Supplementary Figure 21 | Raman spectroscopy. Raman spectroscopy of as-grown MoS₂, the MoS₂-pV3D3 heterostructure, and the MoS₂-Al₂O₃ heterostructure.

(Supplementary Figure 23: in the revised manuscript)

Supplementary Figure 23 | Surface profile of the plano-convex lens. **a**, Surface profile of the plano-convex lens. **b**, Surface roughness of the plano-convex lens. The best fit radius of curvature is 13.127178 mm, the RMS value of the roughness is 0.2798 μm, and the peak-to-peak variation of the roughness is 4.4593 μm.

#4 The first page of Supplementary Information was revised.

(Line 1, page 1: in the revised Supplementary Information)

“Supplementary Information

Curved neuromorphic image sensor array using a MoS₂-organic heterostructure inspired by the

human visual recognition system

Changsoon Choi *et al.*

The PDF file includes:

Supplementary Notes 1-5, Supplementary Figures 1-23, Supplementary Tables 1-4, and Supplementary References.”

#5 Supplementary References were revised.

(Line 12, page 2: in the revised Supplementary Note 1)

“In the conventional imaging device, the complicated multi-lens optics (*e.g.*, double Gauss lens) is required to reduce the spherical aberration, originated from the mismatch between the curved focal plane and the flat image sensor array¹.”

(Line 17, page 2: in the revised Supplementary Note 1)

“In contrast, the curved neuromorphic imaging device, inspired by the human vision, enables aberration-free imaging based on the single-lens optics¹ because the curved neuromorphic image sensor array (cNISA) matches with the curved focal plane of a single plano-convex lens (Supplementary Fig. 3b).”

(Line 9, page 4: in the revised Supplementary Note 3)

“We performed *ab initio* density functional theory (DFT) calculations using the Vienna *Ab initio* simulation package (VASP) to compute the electronic structure of the MoS₂-pV3D3 heterostructure and the MoS₂-Al₂O₃ heterostructure^{3,4}.”

(Line 15, page 4: in the revised Supplementary Note 3)

“The Perdew–Burke–Ernzerhof (PBE) exchange–correlation functional was used⁴.”

(Line 20, page 4: in the revised Supplementary Note 3)

“We used DFT-D3 to apply Gamma-point-centered k-point of $4 \times 4 \times 1$ to the fully relaxed structures for capturing dispersion interactions and computing the density of states and charge differences⁵.”

(Line 2, page 5: in the revised Supplementary Note 3)

“The charge differences were visualized by using the VESTA package⁶. To determine the exciton binding energy, we performed the many-body GW+BSE (Bethe-Salpeter Equation) calculations using the BerkeleyGW package⁷.”

(Line 6, page 5: in the revised Supplementary Note 3)

“The mean field simulations were initiated by using the Quantum Espresso⁸ package based on DFT at the PBE level. We used optimized norm-conserving pseudopotentials⁹ to construct wavefunctions with a plane-wave kinetic energy cutoff of 60 Ry and a Gamma-point-centered k-point grid of $4 \times 4 \times 1$. The dielectric

cutoff was set to 7 Ry.”

(Line 12, page 5: in the revised Supplementary Note 3)

“Such large exciton binding energies could lead to efficient charge transfer at the interface caused by the real-space localization of the electron-hole pairs because the electron-hole pair would find lower energy sites to accommodate the large binding energy, although the efficient charge separation processes overcoming the large exciton binding energy in 2D materials have not been fully understood¹⁰⁻¹².”

(Line 23, page 6: in the revised Supplementary Note 5)

“In Raman spectra of the MoS₂-pV3D3 heterostructure and the MoS₂-Al₂O₃ heterostructure (red line and blue line in Supplementary Fig. 21, respectively), E'_{2g} was barely shifted by pV3D3 and Al₂O₃ deposition, implying the residual strain caused by the deposition process is small¹³. However, A_{1g} was red-shifted in both spectra, implying the deposited dielectric layers induced a slight doping effect to the MoS₂ layer^{14,15}.”

(Line 5, page 7: in the revised Supplementary Note 5)

“PL spectrum of the MoS₂-Al₂O₃ heterostructure exhibits the reduced intensity (blue line in Supplementary Fig. 22a), implying the PL quenching effect by Al₂O₃ layer¹⁶.”

(Line 3, page 36: in the revised Supplementary References)

1. Choi, C. *et al.* Human eye-inspired soft optoelectronic device using high-density MoS₂-graphene curved image sensor array. *Nat. Commun.* **8**, 1664 (2017).
2. Amit, I. *et al.* Role of charge traps in the performance of atomically thin transistors. *Adv. Mater.* **29**, 1605598 (2017).
3. Kresse, G. & Furthmüller, J. Efficient iterative schemes for *ab initio* total-energy calculations using a plane-wave basis set. *Phys. Rev. B* **54**, 11169-11186 (1996).
4. Gaspar, R. & Nagy, A. Local-density-functional approximation for exchange-correlation potential. *Theoretica chimica acta.* **72**, 393-401 (1987).
5. Grimme S., Antony J., Ehrlich S. & Krieg H. A consistent and accurate *ab initio* parametrization of density functional dispersion correction (DFT-D) for the 94 elements H-Pu. *J. Chem. Phys.* **132**, 154104 (2010).
6. Momma K. & Izumi F., VESTA: a three-dimensional visualization system for electronic and structural analysis. *J. Appl. Cryst.* **41**, 653-658 (2008).
7. Deslippe, J., Samsonidze, G., Strubbe, D. A., Jain, M., Cohen, M. L. & Louie, S. G. BerkeleyGW: A massively parallel computer package for the calculation of the quasiparticle and optical properties of materials and nanostructures. *Comput. Phys. Commun.* **183**, 1269-1289 (2012).
8. Giannozzi, P. *et al.* QUANTUM ESPRESSO: a modular and open-source software project for quantum simulations of materials. *J. Phys. Condens.* **21**, 395502 (2009).
9. Hamann, D. R. Optimized norm-conserving Vanderbilt pseudopotentials. *Phys. Rev. B* **88**, 085117 (2013).
10. Massicotte, M. *et al.* Dissociation of two-dimensional excitons in monolayer WSe₂. *Nat. Commun.* **9**, 1633 (2018).
11. Zhu, X. *et al.* Charge transfer excitons at van der Waals interfaces. *J. Am. Chem. Soc.* **137**, 8313-8320 (2015).
12. Zhu, T. *et al.* Highly mobile charge-transfer excitons in two-dimensional WS₂/tetracene heterostructures. *Sci. Adv.* **12**, eaao3104 (2018).
13. Lin, Y. *et al.* Dielectric screening of excitons and trions in single-layer MoS₂. *Nano Lett.* **14**, 5569-5576 (2014).

14. Kukusecka, G. & Koltai, J. Theoretical investigation of strain and doping on the Raman spectra of monolayer MoS₂. *Phys. Status Solidi B* **254**, 1700184 (2017).
15. Cheng, L. *et al.* Sub-10nm tunable hybrid dielectric engineering on MoS₂ for two-dimensional material-based devices. *ACS Nano* **11**, 10243-10252 (2017).
16. Kim, S. Y., Yang, H. I. & Choi, W. Photoluminescence quenching in monolayer transition metal dichalcogenides by Al₂O₃ encapsulation. *Appl. Phys. Lett.* **113**, 133104 (2018).”

#6 The affiliation of authors was changed.

(Line 5, page 1: in the revised main text)

“Changsoon Choi^{1,2†}, Juyoung Leem^{3†}, Min Sung Kim^{1,2†}, Amir Taqieddin^{3,4}, Chullhee Cho³, Kyoung Won Cho^{1,2}, Gil Ju Lee⁴, Hyojin Seung^{1,2}, Hyung Jong Bae³, Young Min Song⁴, Taeghwan Hyeon^{1,2}, Narayana R. Aluru^{3,4}, SungWoo Nam^{3,5*}, and Dae-Hyeong Kim^{1,2,6*}”

(Line 15, page 1: in the revised main text)

~~“⁴Beckman Institute for Advanced Science and Technology, University of Illinois at Urbana-Champaign, Urbana, Illinois 61801, United States.~~

⁴School of Electrical Engineering and Computer Science, Gwangju Institute of Science and Technology, Gwangju 61005, Republic of Korea.

⁵Department of Materials Science and Engineering, University of Illinois at Urbana-Champaign, Urbana, Illinois 61801, United States.

⁶Department of Materials Science and Engineering, Seoul National University, Seoul 08826, Republic of Korea.”

#7 Other minor modification was made for further clarification.

(Line 21, page 17: in the revised main text)

“The data files that support the findings of this study are available from the corresponding authors upon reasonable request.”

(Line 4, page 21: in the revised main text)

“36. Jang, H. *et al.* An atomically thin optoelectronic machine vision processor. *Adv. Mater.* **32**, 2002431 (2020).”

Thank you very much again for your insightful comments. We feel that these comments have helped to improve the quality of the manuscript significantly.